# CropSuite v1.0 - A comprehensive open-source crop suitability model considering climate variability for climate impact assessment

F. Zabel[1], M. Knüttel[1], B. Poschlod[2]

[1]Department of Environmental Sciences, University of Basel, 4056 Basel, Switzerland

[2]Center for Earth System Research and Sustainability, Universität Hamburg, 20144 Hamburg, Germany

*Correspondence to*: florian.zabel@unibas.ch

**Abstract.**

Increasing demand for agricultural land resources and changing climate conditions require for strategic land-use planning and the development of adaptation strategies. Therefore, information about the suitability of agricultural land is a prerequisite. Current suitability approaches often focus on single crops, can only be applied regionally and usually neglect the impact of climate variability on crop suitability. Here, we introduce CropSuite, a new comprehensive and easy-to-use crop suitability model that allows to overcome these shortcomings. It provides a graphical user interface (GUI) and a wide range of pre- and postprocessing options, including a tool for data analysis, which allows users to easily apply the model and analyze the results. Further, it includes a spatial downscaling approach for climate data, which enables crop suitability analysis at very high spatial resolution. CropSuite uses a fuzzy logic approach and is based on the assumption of Liebig's law of the minimum. An expandable number of environmental and socio-economic factors that impact on crop suitability can flexibly be integrated into CropSuite by determining membership functions. CropSuite allows for the consideration of irrigated and rainfed agricultural systems, vernalization requirements for winter crops, lethal temperature thresholds, photoperiodic sensitivity and several other limitations for crop growth. The model endogenously calculates and outputs climate-, soil-, and crop suitability, the optimal sowing- and harvest dates, the potential for multiple cropping, the (most) limiting factor(s), as well as the recurrence rate of potential crop failures according to the inter-annual climate variability.

In this study, we apply CropSuite for 48 crops at a spatial resolution of 30 arc seconds (1 km at the equator) for Africa. Thereby, we consider regionally important staple and cash crops that are usually understudied, such as coffee, cassava, banana, oil palm, cocoa, cowpea, groundnuts, mango, millet, papaya, rubber, sesame, sorghum, sugar cane, tobacco, and yams. We find that the consideration of climate variability for calculating crop suitability makes a significant difference on suitable areas, but also affects optimal sowing dates, and multiple cropping potentials. The most vulnerable regions

for climate variability are identified in Somalia, Kenya, Ethiopia, South Africa, and the Maghreb countries. The results
provide valuable crop-specific information that can be further used for climate impact assessments, adaptation and land-
use planning at global, regional, or local scale. CropSuite is provided open source and could be of interest for model
developers, scientists, and a wide range of potential users and stakeholders, such as farmers, companies, GOs, and NGOs.

**Key Words: Agriculture, Africa, Optimal Sowing Dates, Multiple Cropping, Maize**
**1 Introduction**
Climate change poses major challenges for agricultural production and food security. With warming climate, agricultural
suitability changes and suitable areas shift towards higher latitudes (Franke et al., 2021; Zabel et al., 2014). Crop
suitability models allow for a quantitative evaluation of land for crop cultivation and can therefore assess how the
suitability of land changes with changing climate. Contrary to mechanistic crop models (Jägermeyr et al., 2021;
Jägermeyr et al., 2020; Müller et al., 2024), crop suitability models are based on empirical approaches but are less
computational intensive and thus allow for the consideration of more crops at higher spatial resolution (Zabel et al., 2014).
As a result, crop suitability models provide important insights for sustainable land-use planning and climate change
adaptation, e.g. through cultivar change or land-use change. Akpoti et al. (2019) give an overview of existing crop
suitability approaches. Most studies are applied at regional scale (Maleki et al., 2017; Bonfante et al., 2015; Ranjitkar et
al., 2016), while just a few global approaches exist (Akpoti et al., 2019). In addition, most studies focus just on single
crops and do not cover a variety of different crops (Ramirez-Villegas et al., 2013; Akpoti et al., 2020). Particularly for
Africa, domestically consumed staple crops, such as yams and cassava are often overseen in current studies, due to minor
economic relevance, despite their regional importance for food security (Chapman et al., 2020; Chemura et al., 2024;
Van Zonneveld et al., 2023; Karl et al., 2024). So far, none of the existing approaches systematically considers the impact
of climate variability on crop suitability, which is a major shortcoming, since climate variability is expected to increase
with climate warming and has a strong impact on agriculture (Vogel et al., 2019; Goulart et al., 2021; Ipcc, 2021).
The aim of this study is to introduce the CropSuite model, which is based on the crop suitability approach developed by
Zabel et al. (2014) and has continuously been further developed by Cronin et al. (2020) and Schneider et al. (2022a). The
model has previously been applied globally for 23 crops for different climate scenarios (Zabel, 2022). The model applies
Liebig's law of the minimum, assuming that the scarcest resource limits the crop growth. CropSuite is based on a fuzzy
logic approach where, in contrast to Boolean logic, the truth value of variables can be any real number between 0 and 1.
In fuzzy logic, fuzzy sets consist of elements whose degrees of memberships are described by membership functions
(Zadeh L.A., 1965). In our approach, we apply fuzzy logic to create crop-specific membership functions (Fig. 1)
describing the abiotic crop requirements between 0 (not suitable) and 100 (highly suitable) according to various climatic,
soil, and topographic variables (Zabel et al., 2014). Using a value range between 0 and 100 (instead of 0 and 1) enables

the use of an 8-bit integer data type for the internal calculation and storage of the results, which allows efficient use of memory and hard disk. This approach is adopted, fundamentally redesigned and expanded with the goal to provide a comprehensive but easy-to-use and flexible open-source model that can be applied e.g. by scientists, farmers, companies, national or international GOs, and NGOs. Therefore, CropSuite is now completely reprogrammed in Python and consists of a graphical user interface (GUI), as well as several pre-processing and analysis tools, e.g. for selecting a simulation domain, statistically downscaling the climate data, interpolating the membership functions and automatically analyzing and mapping the results. In addition, CropSuite is complemented with a new approach to consider the impact of climate variability on crop suitability. It includes a user manual, which is provided together with the source code (Knüttel and Zabel, 2024).

## 2 Methods and Data

For this study, we apply CropSuite for Africa at 30 arc seconds spatial resolution (approximately 1 km$^2$ at the equator) with the goal to simulate relevant but often overseen crops for this continent (Van Zonneveld et al., 2023). Table 1 shows the 48 crops, that have been parameterized and simulated with CropSuite.

**Table 1: List of 48 considered crops simulated with CropSuite.** Binomial names are given in brackets.

| | |
|---|---|
| 1. Alfalfa *(Medicago sativa)* | 25. Olive *(Olea europacae)* |
| 2. Arabica Coffee *(Coffea arabica)* | 26. Onion *(Allium cepa)* |
| 3. Avocado *(Persea americana)* | 27. Papaya *(Carica papaya)* |
| 4. Banana *(Musea spp.)* | 28. Pea *(Pisum sativum)* |
| 5. Barley *(Hordeum vulgare)* | 29. Pineapple *(Ananas comosus)* |
| 6. Beans *(Phaseolus vulgaris)* | 30. Potato *(Solanum tuberosum)* |
| 7. Cabbage *(Brassica oleracca)* | 31. Rapeseed *(Brassica napus)* |
| 8. Carrot *(Daucus carota)* | 32. Rice *(Oryza sativa)* |
| 9. Cashew *(Anacardium occidentale)* | 33. Robusta Coffee *(Coffea canephora)* |
| 10. Cassava *(Manihot esculenta)* | 34. Rubber *(Hevea brasiliensis)* |
| 11. Castor Bean *(Ricinus commuis)* | 35. Rye *(Secale cereale)* |
| 12. Chickpea *(Cicer arietinum)* | 36. Safflower *(Carthamus tinctorius)* |
| 13. Citrus *(Citrus spp.)* | 37. Sesame *(Sesamum indicum)* |
| 14. Cocoa *(Theobroma cacao)* | 38. Sorghum *(Sorghum bicolor)* |
| 15. Coconut *(Cocos nucifera)* | 39. Soy *(Glycine maximum)* |
| 16. Cotton *(Gossypium hirsutum)* | 40. Sugar Cane *(Saccharum officinarum)* |
| 17. Cowpea *(Vigna unguiculata)* | 41. Sunflower *(Helianthus annus)* |
| 18. Green Pepper *(Capsium annuum)* | 42. Sweet Potato *(Ipomoea batatas)* |
| 19. Groundut *(Arachis hypogaea)* | 43. Tea *(Camellia senesis)* |
| 20. Guava *(Psidium guijava)* | 44. Tobacco *(Nicotiana tabacum)* |
| 21. Maize *(Zea mais)* | 45. Tomato *(Solanum lycopersicum esculentum)* |
| 22. Mango *(Mangifera indica)* | 46. Watermelon *(Colocynthis citrullus)* |

| 23. Millet *(Pennisetum americanum)* | 47. Wheat *(Triticum aesticum)* |
| 24. Oil Palm *(Elaeis guineensis)* | 48. Yams *(Dioscorea)* |

We simulate a 20-year time period from 1991 to 2010 using the Climate Hazards group Infrared Precipitation with Stations (CHIRPS) v2.0 daily data for precipitation (Funk et al., 2015) and the Climate Hazards Center Infrared Temperature with Stations (CHIRTS) v1.0 data for temperature (Funk et al., 2019; Verdin et al., 2020) at 2.5 arc minutes spatial resolution for Africa. Both data sets provide climatologies at daily to monthly resolution based on a combination of satellite remote sensing and climate stations. They benefit from long-term geostationary satellite observations, delivering consistent data since the 1980s at the quasi-global (50°S-50°N) scale.

In addition, soil and terrain information is required. Table 2 gives an overview of the soil and terrain data used for this study. Soil data is mainly based on ISRIC SoilGrids (Hengl et al., 2017), which has a spatial resolution of 250 m but is also provided at 1000 m spatial resolution. This data is reprojected to WGS84 and spatially interpolated using nearest neighbor to the spatial resolution of 30 arc seconds applied in this study. Base saturation, gypsum, and exchangeable sodium content (ESP, sodicity) are taken from the WISE database at a spatial resolution of 30 arc seconds (Batjes, 2016). For electric conductivity, the ISRIC Global Soil Salinity Map with a resolution of 250 m is used (Ivushkin et al., 2019). In contrast to the harmonized world soil database (HWSD) (Fao et al., 2012), the ISRIC soil datasets do not contain a layer for texture class. For this reason, the texture class is determined using the sand and clay layer of SoilGrids according to the United States Department of Agriculture (USDA) triangular diagram of soil texture classes (Fao et al., 2012). For soil depths greater than 200 cm up to 50 m, the ISRIC dataset on absolute depth to bedrock (Hengl et al., 2017) is complemented with the dataset from Pelletier et al. (2016), which covers soil depths up to 200 cm.

Available soil layers can be weighted in CropSuite as required. The SoilGrids datasets provide information for six depths: 0-5 cm, 5-15 cm, 15-30 cm, 30-60 cm, 60-100 cm, and 100-200 cm (Hengl et al., 2017; Hengl et al., 2014). According to Sys et al. (1991), soil properties have different effects on crop suitability depending on the soil layer. Accordingly, we use weighting factors as proposed by Sys et al. (1991) (see Table 2). The different distribution of the soil depths between the SoilGrids data and the weighting factors by Sys et al. (1991) is taken into account by using a proportional weighting of the SoilGrids layers.Terrain data are taken from the Shuttle Radar Topography Mission (SRTM) data set (Farr et al., 2007), which are used to calculate the slope at the applied spatial resolution. Please be aware that a coarser spatial resolution generally reduces the slope, which could result in an underestimation of possible slope limitations in mountainous regions. A possible terracing could remove the restriction due to the slope but usually terraces are too small to be considered at the aggregated spatial resolution of 30 arc seconds of the SRTM data in this study.

**Table 2: Soil and terrain data used in this study and the applied weighting of the different soil layers.**

| Parameter | Source | Weighting |
|---|---|---|
| Base Saturation | ISRIC Harmonized Dataset of Derived Soil Properties for the World (WISE30sec) (Batjes, | Only Top Soil |

| | 2016) | |
|---|---|---|
| Coarse Fragments | ISRIC SoilGrids 250m (Hengl et al., 2017) | 0 - 25 cm: 2.0<br>25 - 50 cm: 1.5<br>50 - 75 cm: 1.0<br>75 - 100 cm: 0.75<br>100 - 125 cm: 0.5<br>125 - 150 cm: 0.25 |
| Electric Conductivity | ISRIC Global Soil Salinity Map (Ivushkin et al., 2019) | Only Top Soil |
| Gypsum Content | ISRIC Harmonized Dataset of Derived Soil Properties for the World (WISE30sec) (Batjes, 2016) | Only Top Soil |
| Organic Carbon Content | ISRIC SoilGrids 250m (Hengl et al., 2017) | 0 - 25 cm: 2.0<br>25 - 50 cm: 1.5<br>50 - 75 cm: 1.0<br>75 - 100 cm: 0.75<br>100 - 125 cm: 0.5<br>125 - 150 cm: 0.25 |
| Soil pH | ISRIC SoilGrids 250m (Hengl et al., 2017) | 0 - 5 cm: 0.33<br>5 - 15 cm: 0.33<br>15 - 30 cm: 0.33 |
| Sodicity | ISRIC Harmonized Dataset of Derived Soil Properties for the World (WISE30sec) (Batjes, 2016) | Only Top Soil |
| Soil Depth | ISRIC SoilGrids 2017 (Soil Depth <= 200 cm) (Hengl et al., 2017)<br><br>Pelletier et al. (2016) (Soil Depth > 200 cm) | No Weighting |
| Texture Class | Texture class calculated from ISRIC SoilGrids 250m clay and sand content (Hengl et al., 2017) according to USDA (Fao et al., 2012) | 0 - 25 cm: 2.0<br>25 - 50 cm: 1.5<br>50 - 75 cm: 1.0<br>75 - 100 cm: 0.75<br>100 - 125 cm: 0.5<br>125 - 150 cm: 0.25 |
| Slope | SRTM aggregated to 30 arcsec (Farr et al., 2007) | No Weighting |


Membership functions for temperature, precipitation, slope, soil depth, texture class, coarse fragments, gypsum, base
saturation, pH, organic carbon, electric conductivity, sodicity (Fig. 1) are defined for the considered 48 crops relying on
information from Sys et al. (1993), which provide membership functions for most of the considered crops. Additionally,
data from the EcoCrop database, which provides crop ecological requirements for more than 2500 plant species (Fao,
2024), is used for Cowpea, Rye, and Yams. CropSuite in principle allows the flexible addition of any further membership
function and dataset that is relevant for the use case.
Nutrient deficits, such as nitrogen content are not considered in our approach, since according to our definition of crop
suitability, they are not a decisive factor for the suitability of crops but rather depend on the crop management.
Accordingly, we do not consider any soil tillage that can affect the soil properties, such as liming, which can influence
the pH value.

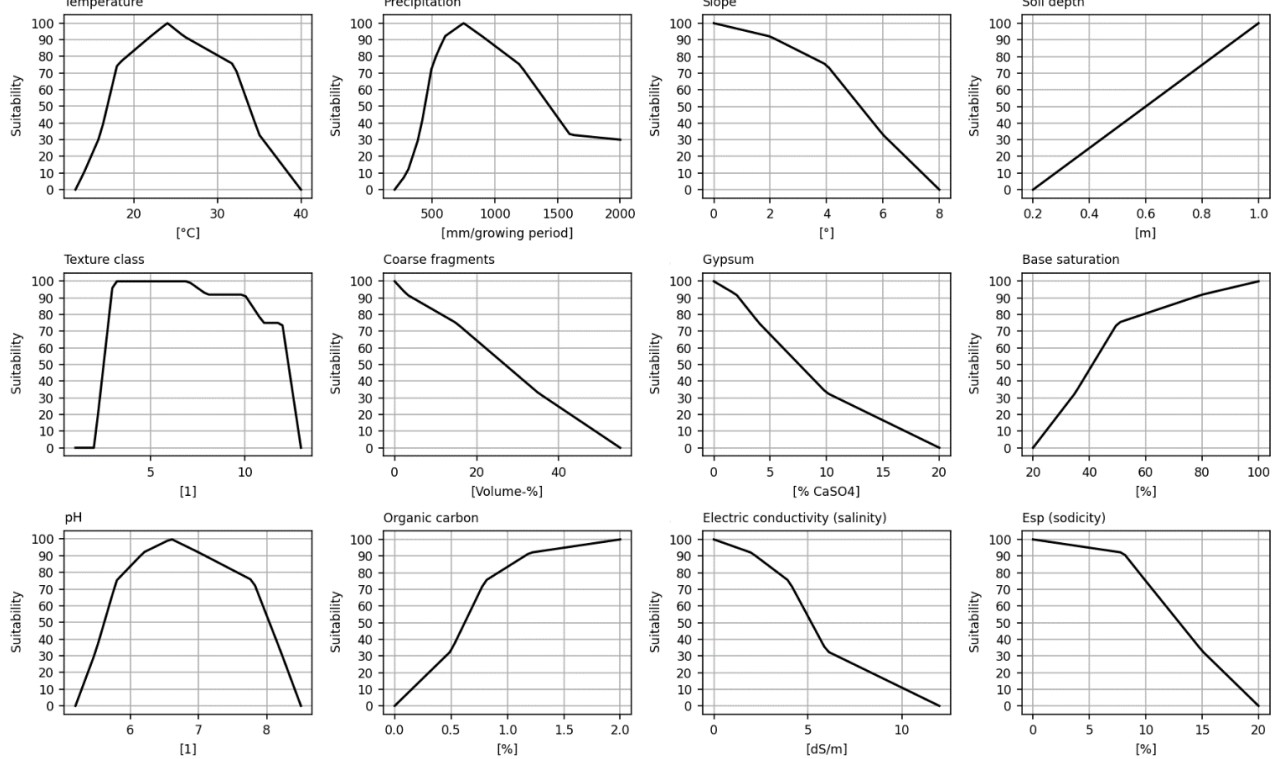


**Figure 1: Membership functions exemplarily for maize** with a growing cycle of 110 days for considered climatic (mean temperature
over the growing cycle, total precipitation over the growing cycle), topographic (slope), and soil constraints (soil depth, texture class,
coarse fragments, gypsum, base saturation, pH, organic carbon, salinity, sodicity).
Sys et al. (1993) uses a classification system with 6 classes, ranging from N2 as unsuitable to S0 as highly suitable. In
this study, we dismiss the N1 class due to a vague definition and differentiate three suitability classes, marginally,
moderately, and highly suitable (Table 3).

**Table 3: Crop suitability classification system as used in this study compared to Sys et al. (1993).**

| Suitability classes according to Sys et al. | Suitability range | Suitability classes used in this study |
|---|---|---|
| S0   (highly suitable) | 100 | 75 – 100   (highly suitable) |
| S1   (very suitable) | 80 – 99 | |
| S2   (moderately suitable) | 60 – 79 | 33 – 74      (moderately suitable) |
| S3   (marginally suitable) | 40 – 59 | 1 – 32       (marginally suitable) |
| N1   (actually unsuitable and potentially suitable) | 20 – 39 | 0             (unsuitable) |
| N2   (unsuitable) | 0 - 19 | |

## 2.1 The CropSuite Model

Figure 2 shows the workflow and outputs of CropSuite, which first calculates a climate suitability (considering all climate constraints) and then calculates a soil suitability (considering all soil and topography constraints). Both data records can be output separately. Thereby, CropSuite applies Liebig's law of the minimum, for both the climate and the soil suitability by choosing the lowest suitability value between the different soil parameters and climate variables respectively. Finally, the crop suitability is calculated from the combination of both climate and soil suitability by again following Liebig's law of the minimum, which means that the lowest suitability value between climate and soil suitability is chosen, since it restricts overall crop suitability. The most limiting factor is identified as the parameter that imposes the greatest constraint on growth for a specific crop. In addition, the magnitude of the constraint is output for each input factor. Overall, CropSuite allows for a variety of outputs on optimal sowing- and harvest dates, suitable sowing days, multiple cropping potentials, the limiting factor, and the recurrence rate of potential crop failures. Output data format can be set to GeoTIFF or NetCDF.

CropSuite includes a pre-processing procedure which creates intermediate results for climate variability. Since climate model data are usually available at relatively coarse spatial resolution, CropSuite has implemented a spatial downscaling module for the climate data, which allows the model to be applied at very high spatial resolution from global to regional to local scale. In this study, we apply a statistical downscaling to the climate data, refining the spatial resolution from 2.5 arc minutes to 30 arc seconds. In principle, the targeted spatial resolution can be set in CropSuite but is limited to the available resolution of the additional input data, such as the soil data, whereas for the climate data, two different statistical spatial downscaling methods are implemented requiring little computational effort. The first methodology is based on an altitude regression for temperature (Marke et al., 2014), where the temperature gradients are extracted from the climate model data itself via a moving window that can be set in size. Thereby, the extracted gradients must remain within the natural boundaries for wet and dry adiabatic temperature gradients. The second downscaling methodology uses the historical high-resolution spatial patterns for monthly temperature and precipitation taken from WorldClim at 30 arc seconds spatial resolution (Fick and Hijmans, 2017). To downscale a coarse-resolution grid cell, all fine-resolution WorldClim grid cells within the coarse-resolution cell are selected and aggregated per month. On this basis, additive factors are calculated for temperature and multiplicative factors for precipitation separately for each month. Thereby the sum (mean) of these additive (multiplicative) factors within the coarse-resolution cell amounts 0 (1). Considering the monthly seasonality, these factors are applied to the coarse-resolution climate data, imprinting the spatial pattern of the high-resolution reference data onto the coarse climate data at daily time step. Both downscaling methods conserve mass and energy from the climate input data by iteratively minimizing residuals over the simulation domain. For a more advanced statistical downscaling to kilometer-scale, the expert user may apply more complex topographical downscaling methods (Daly et al., 1994; Fiddes et al., 2022; Karger et al., 2023) or downscaling based on machine learning (Damiani et al., 2024; Wang et al., 2021) outside of CropSuite. Furthermore, we do not recommend applying the implemented

downscaling methods with high scaling factors from very coarse (hundreds of kilometers) to very high (single kilometer)
resolution.

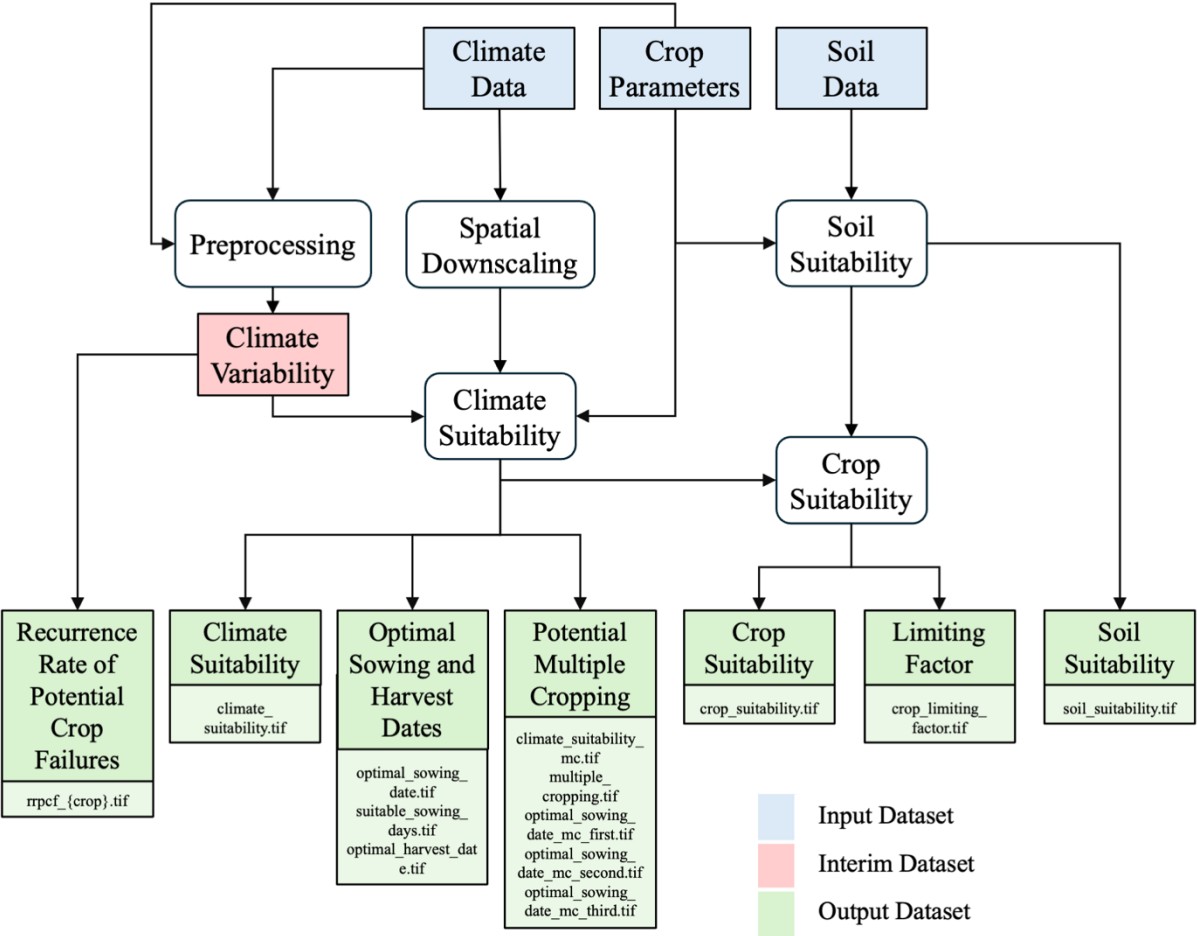

**Figure 2: CropSuite workflow.** Input data in blue, intermediate results in red and output data in green. The processing steps are
shown in white.
CropSuite requires daily climate data as an input for temperature and precipitation. As climate models tend to produce
too many days with low-intensity precipitation called "drizzle bias" (Chen et al., 2021), days with aggregated daily
precipitation values below 1 mm per day are considered to be dry days (Sun et al., 2006). This threshold can be set in the
model. Both downscaled temperature and precipitation data and the calculated datasets for climate variability are used to
calculate the climate suitability. Therefore, the crop-specific membership functions determine the suitability according
to the average temperature, total precipitation and the recurrence rate of potential crop failures over the length of the
growing cycle (time from sowing till maturity) for each day of year (DOY). Thereby, the suitability value for each DOY
refers to the average conditions during the growing cycle from that DOY, which corresponds to the sowing date, until

maturity, determined by the length of the growing cycle which is set in the crop parameterization for each crop. For perennial crops, the length of the growing cycle is set to 365 days. Climate suitability throughout the year is then identified by selecting the minimum value (most limiting) of the three individual suitabilities for temperature, precipitation, and climate variability. As shown in Fig. 3, the DOY with the highest climate suitability value over the year finally determines the optimal sowing date for annual crops (optimal planting date for rice, which is not sown, but planted as a seedling in wet rice cultivation). For perennial crops this is set to 1.

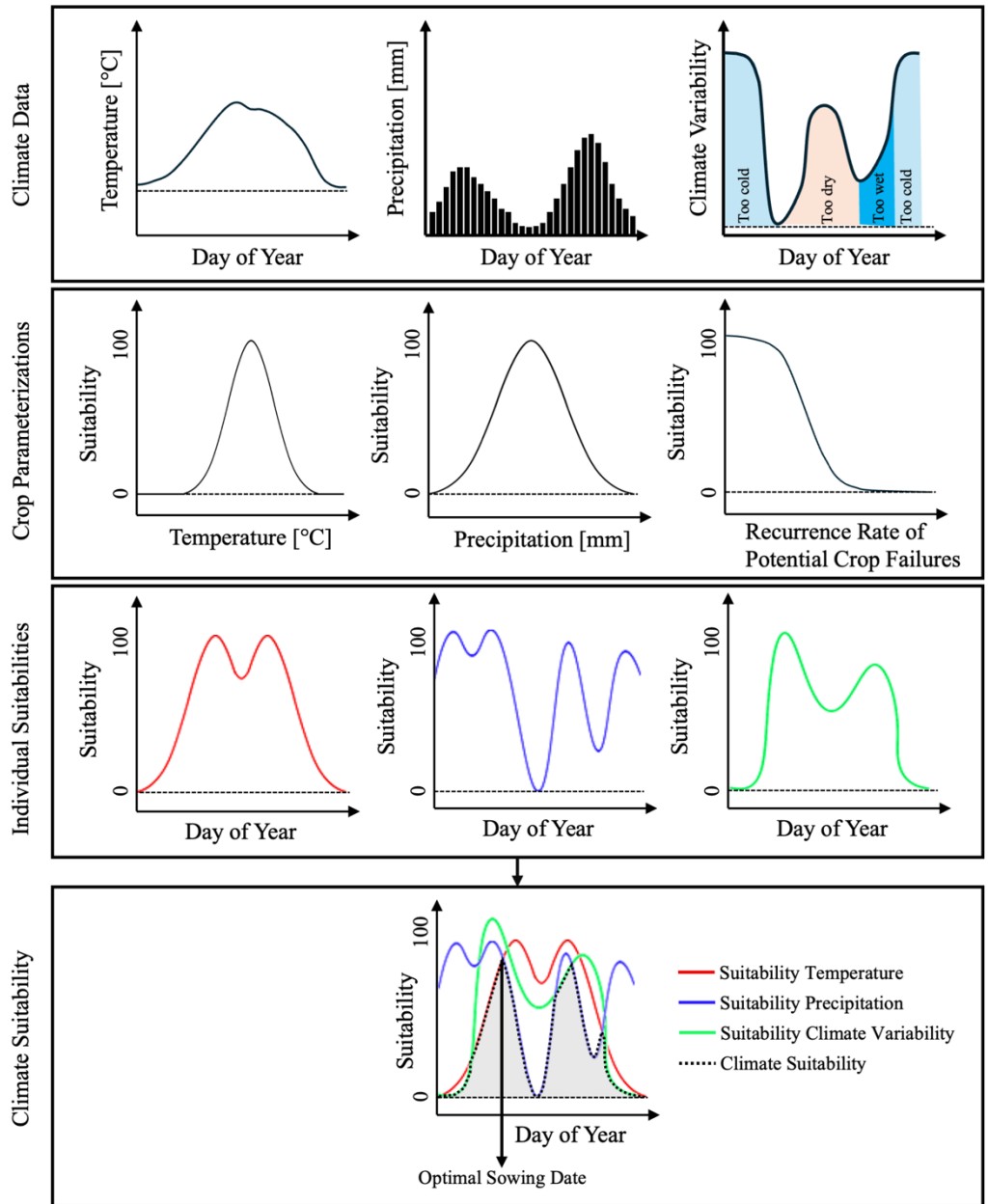

180

**Figure 3: Schematic illustration of the determination of climate suitability, the optimal sowing date and the limiting factor.** The input data shows the annual course of temperature, precipitation and the recurrence rate of potential crop failure, indicating whether it is too cold, too dry, or too wet. The crop parameterizations show the membership functions resulting in the individual suitability values for each DOY for either temperature (red line), precipitation (blue line), and climate variability (green line).Climate suitability throughout the year (black dashed line) results from the lowest of the three curves (most limiting) on any day. The highest value of climate suitability over the year finally determines the optimal sowing date. The limiting factor is the most constraining factor at this point.

For annual crops, CropSuite also calculates the potential for multiple harvests without considering crop rotation. Between
harvest and reseeding, we assume a certain time period (21 days in this study) for field work and processing, which can
be set flexibly in the model. Accordingly, all possible combinations of sowing dates are tested with the aim to maximize
climatic suitability to achieve the highest sum of climatic suitability within a year. The optimal sowing dates are selected
from the best sowing date combinations, resulting in one, two, or three sowing dates per year. A multiple cropping layer
is output that shows how often a crop can be harvested.
CropSuite distinguishes between rainfed and irrigated agricultural systems, which can be selected before starting the
simulation. For the irrigated case, precipitation is not considered as a constraining factor with consequences for all further
calculations, affecting e.g. the climate variability, the optimal sowing date, and the multiple cropping. For this study, we
separately simulated both, rainfed and irrigated options for all crops. In the post-processing, we combined both datasets
according to the irrigated areas dataset by Meier et al. (2018) (Fig. S1), which is available at 30 arc-seconds spatial
resolution.
For germination, crop-specific temperature and soil water requirements can be set in the model. The latter can be
considered for rainfed conditions by defining a certain amount of precipitation within a certain period of time after
sowing.
Some crops, such as soybean have a high photoperiodic sensitivity which can limit their suitability (Cober and Morrison,
2010; Abdulai et al., 2012). Therefore, crop-specific photoperiodic sensitivity can be considered in CropSuite by defining
a maximum and minimum day length in average over the growing cycle.
Additional lethal climatic limitations can be taken into account in CropSuite. We assume permafrost on areas with an
average annual temperature below 0° C, which is computed from the downscaled climate input data. A maximum lethal
temperature threshold of >40°C in average over the growing cycle is set for all crops (Asseng et al., 2021). In addition, a
minimum and maximum threshold for the lethal temperature over a certain consecutive number of days can be set in the
model crop-specifically. Further, the maximum number of consecutive dry days can be set dependent on the
crop.CropSuite allows for the consideration of vernalization requirements for winter crops. Therefore, crop-specific
temperature requirements with minimal and maximal temperature thresholds for a certain number of vernalization
effective days can be configured in the model. Accordingly, CropSuite simulates for each location, if and when these
vernalization requirements are fulfilled, which impacts on the length of the vernalization period and the optimal sowing
date. An offset of days from sowing to the start of the vernalization period can optionally be added.
A GUI is available for CropSuite that allows users to easily set-up the model, parameterize the crop requirements and the
membership functions (Fig. 4a-e), and to start the simulations. Further, new membership functions can be created, an
unlimited number of crop-specific requirements can be defined, and any additional data can be added, which can be
flexibly assigned to the defined membership functions (Fig. 4e). Moreover, new crops or crop varieties can be added.
The GUI also allows for the visualization, analysis and comparison of the results (Fig. 4f).

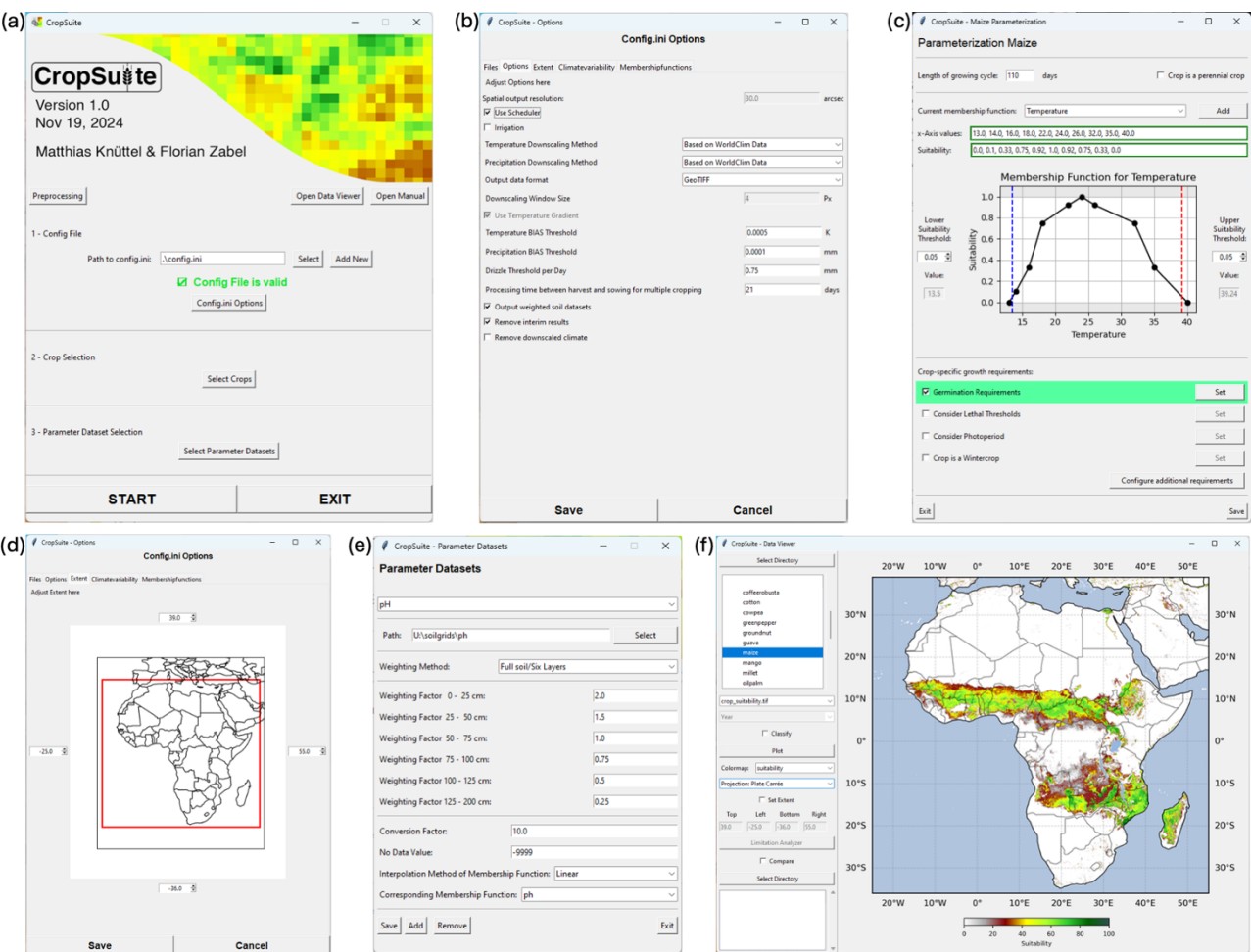

**Figure 4: Graphical User Interface of CropSuite.** (a) shows the main screen, (b) exemplarily shows available model settings, (c) shows the available options for crop parameterizations exemplarily for maize, (d) shows the window to set-up the simulation domain, (e) exemplarily shows the set-up of a parameter dataset for soil pH, and (f) shows the integrated data viewer in CropSuite.

## 2.2 Climate Variability

In addition to several improvements and redesigns, one of the most important advancements in CropSuite is the consideration of climate variability for the assessment of crop suitability. Usually, crop suitability models consider long-term climate averages, e.g. 10, 20 or 30-year periods and climatic trends that affect crop suitability (Ramirez-Villegas et al., 2013; Schneider et al., 2022b). They are not designed so simulate seasonal yields, as for instants mechanistic crop models do (Jägermeyr et al., 2021). However, existing crop suitability approaches may overestimate crop suitability when only long-term averages are considered, because a high climatic variability may result in a high frequency of unsuitable years, which would result in crop failures. This would however significantly increase the risk for farmers that require stable and plannable conditions. As a result, a farmer may conclude that the risk of crop failures due to unstable climate

conditions in a certain region is too high for being suitable for crop cultivation. As such, climate variability is not a purely
ecological limitation but depends on the socio-economic circumstances of how farmers deal with the risk of crop failure.
We developed an approach that allows for the consideration of climate variability, and thus the implicit integration of
socio-economic limitations in the suitability assessment of crops.
Therefore, we specify a crop-specific lower and upper threshold for temperature and precipitation. We recommend these
thresholds between the higher and lower 5% and 10% suitability values of the crop-specific membership function,
respectively (Figs. 1, 4c). If the suitability of the membership function does not approach 0 at its high (low) limit, we
recommend setting the threshold to the highest (lowest) value of the membership function. This is the case for the wet
limit of the precipitation membership function for maize (see Fig. 4c). For each year within a given period of time (here
we use 20-year time periods), it is tested and totaled, how often these thresholds exceed or fall below during the growing
cycle for all possible sowing dates (January 1$^{st}$ until December 31$^{st}$). As a result, a variability dataset is generated for each
DOY, indicating the number of years in which at least either the temperature or the precipitation exceeds or falls below
the threshold values. The number of years is divided by the length of the time period (here 20 years) to obtain the
recurrence rate of potential crop failures. This data can be stored as a two-dimensional raster file for perennial crops or
as a three-dimensional raster file for non-perennial crops, with each of the 365 DOYs representing the condition for the
respective sowing day.
For rainfed agricultural systems, cases that are considered for climate variability include excessively high or low
temperatures and precipitation, while for irrigated agricultural systems, only excessively high or low temperatures and
excessively high precipitation are considered, to address potential water logging, plant diseases or root rotting. Due to
computational limitations, the preprocessing of the climate variability is carried out at the resolution of the input climate
data (2.5 arc minutes) and is further interpolated bilinearly to the output resolution of 30 arc seconds.
Finally, we introduce a membership function defining the impact of climate variability on crop suitability. As shown in
Fig. 5, a sigmoid is adopted for the course of the function. According to expert knowledge, we set this sigmoid function
in a way that it reduces suitability to 0 when the recurrence rate of potential crop failure is greater than once every 4 years
(25%). However, this function may be different in different parts of the world and different between crops (see
Discussion).

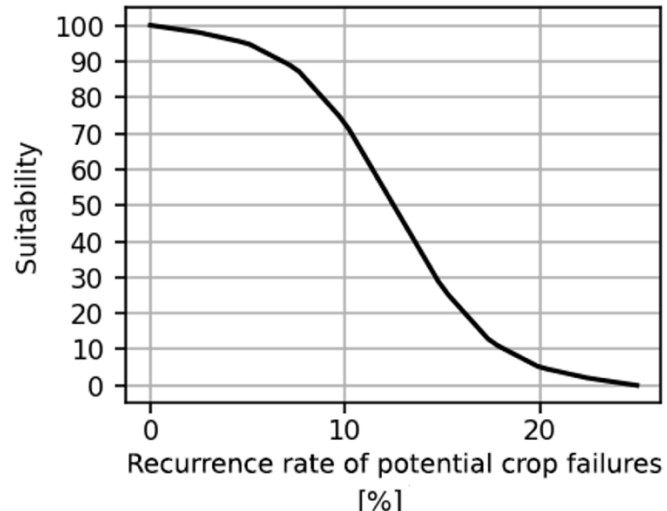


**Figure 5: Membership function for climate variability showing the impact of the recurrence rate of potential crop failures on crop suitability.** The seasonal recurrence rate is shown in percent.

**3 Model evaluation**

Crop suitability is difficult to validate or measure, nor is it equivalent to agricultural yields or production values. However, a comparison with other studies and data can provide valuable information and build confidence in the approach.

**3.1 Comparison with Harvested Area**

In principle, a crop should be suitable where it is already cultivated. According to this premise, we compare the suitable area simulated with CropSuite with the harvested areas from the global spatially-disaggregated crop production statistics data for 2020 (MapSPAM 2020 v1.0) produced by the International Food Policy Research Institute (IFPRI) using the Spatial Production Allocation Model (SPAM) (Ifpri, 2024). The CropSuite results for Africa consider climate variability and are combined for irrigated and rainfed areas according to Meier et al. (2018). While MapSPAM relates to the year 2020, our simulations refer to the 1991-2010 time period, which could be a source of uncertainty. Nevertheless, we used MapSPAM 2020 instead of other available versions of MapSPAM, since it includes 32 crops from our investigation and is the latest released version of MapSPAM. A comparison between CropSuite and different MapSPAM versions is shown exemplarily for maize in Fig. S2, revealing a considerably better fit with CropSuite in the MapSPAM 2020 version. For comparison, harvested areas below 10 ha per pixel are excluded from the calculation and the high spatial resolution of the CropSuite model output is resampled to the same spatial resolution (5 arc minutes) than the MapSPAM 2020 data. Figure 6 depicts the results of this analysis for all crops, where green and purple bars represent areas that are suitable, while orange and green areas represent harvested areas in MapSPAM. Purple bars indicate suitable areas that are currently not used by the respective crop.While green areas are also identified as being suitable in our approach, orange areas are

not suitable in CropSuite despite the respective crop is harvested according to MapSPAM. Crops with the largest
mismatching areas are rice, maize, and onion (Fig. 6). Most crops show a small proportion of orange to green areas,
except for onions, rapeseed, cocoa, pea, rubber, tea, coffee, and rice (Fig. S3). This can have various causes, such as data
uncertainty of climate, soil and irrigation data (Avellan et al., 2012), incorrect membership functions, the use of different
crop varieties, or an incorrect localization of the cultivation areas in MapSPAM due to high uncertainties in the underlying
national statistical data, especially in African countries (Yu et al., 2020), or applied crop management practices that could
level out ecological limitations.

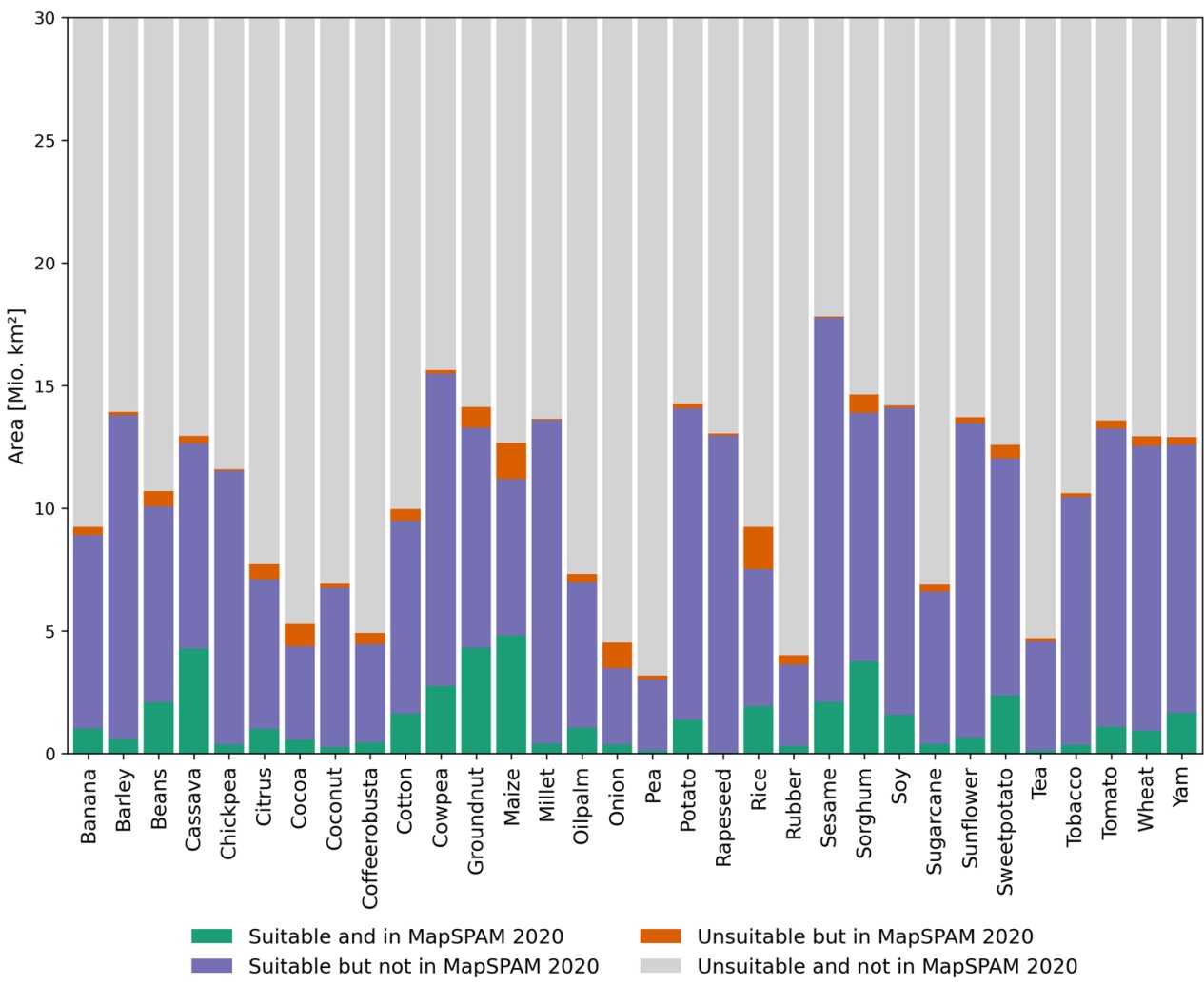


**Figure 6: Comparison of CropSuite with MapSPAM 2020 for all matching crops.** CropSuite results combine irrigated and rainfed
areas according to Meier et al. (2018) and consider climate variability. Areas on which the respective crop is harvested according to
MapSPAM and which are suitable according to CropSuite are shown in green, areas that are suitable but on which the crop is not
harvested are shown in purple. Areas that are unsuitable but are harvested according to MapSPAM are shown in orange, while
unsuitable areas that are not harvested according to MapSPAM are shown in gray.

Figure 7a shows the spatial comparison between crop suitability and harvested areas for maize. Areas where maize is
harvested according to MapSPAM, although CropSuite has identified these areas as unsuitable, are found mainly in
Egypt, the northern Sahel, the Congo Basin, as well as parts of Cameroon, Gabon, Kenya, Tanzania, Zimbabwe and
South Africa. Figure 7b shows the comparison ignoring the impact of climate variability on crop suitability. Disregarding
climate variability results in large (blue) areas, which are considered suitable but are no harvest areas according to
MapSPAM, especially along the dry belts (15°N and 20°S). Our approach considering climate variability (Fig. 7a)
reduces these blue areas, but induces some mismatches, where MapSPAM indicates harvested areas and CropSuite shows
no suitability (red areas). We find that the mismatching areas along the dry belts (including the Sahel) and in eastern
Africa (Tanzania, Kenya) are often associated with limits due to climate variability. This indicates that the thresholds for
climate variability (section 2.2) and the membership function (Fig. 5) might be parameterized slightly too exclusive.
However, some of these regions might be used as cropland by smallholders or subsistence farmers despite the high risk
of crop failures.
While in the inner tropics, the reason for limited crop suitability can primarily be attributed to soil acidity (pH), indicating
possible uncertainties with used SoilGrids dataset, differences in Egypt mainly result from discrepancies according to
different assumptions on irrigated areas.

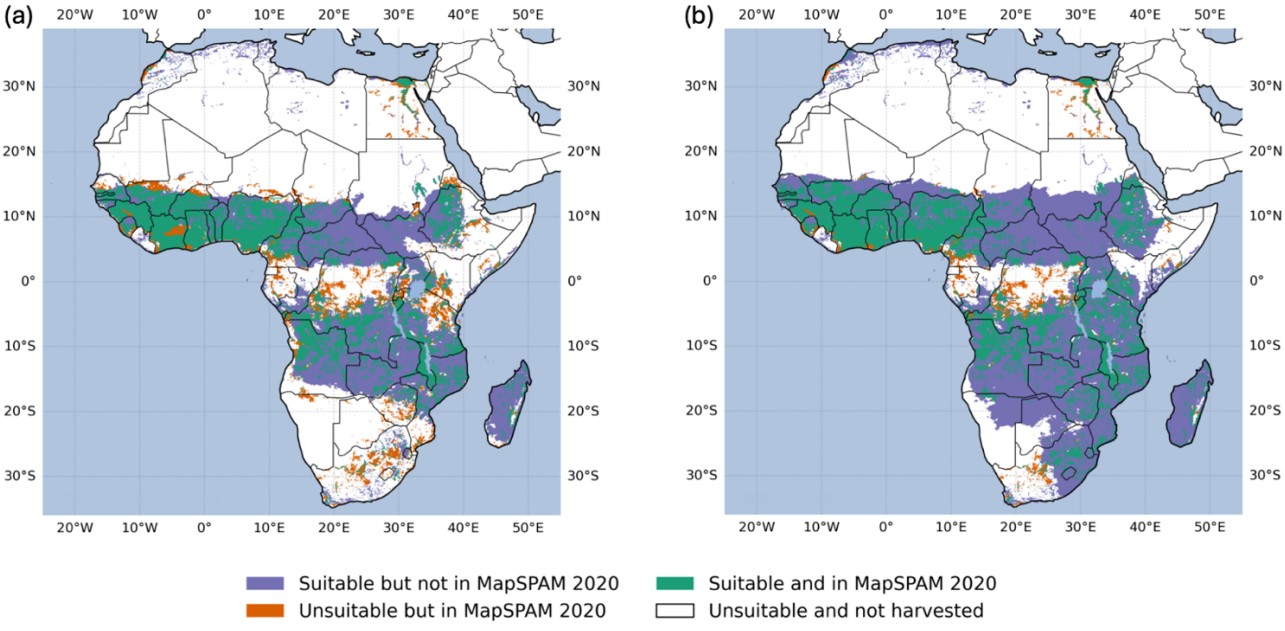

**Figure 7: Comparison of CropSuite with MapSPAM 2020 for maize.** (a) shows the comparison with consideration of climate
variability in CropSuite, while climate variability is not considered in (b). Areas on which the respective crop is harvested according
to MapSPAM and which are suitable according to CropSuite are shown in green, areas that are suitable but on which the crop is not
harvested are shown in blue. Areas that are not suitable but are harvested according to MapSPAM are shown in red. Unsuitable areas
that are not harvested according to MapSPAM are shown in white.

## 3.2 Comparison with GAEZ

A state-of-the-art climate-edaphic suitability assessment for crops is provided by the Global Agro-Ecological Zones (GAEZ) v4 (Fischer et al., 2021). For comparison with CropSuite, we used GAEZ data for the time period 1981-2010 for high input level, rainfed conditions and the option 'all land in grid cell'. The high input level refers to advanced management assumptions (fully mechanized, optimum application of nutrients and chemical pest, disease and weed control) (Fischer et al., 2021), which correspond best to the assumptions made in CropSuite for this study. The suitability range of the GAEZ data is transformed to the classification system as shown in Table 3. The CropSuite data for rainfed conditions is resampled (using the average) to the same spatial resolution of 5 arc minutes than the GAEZ data. For this comparison, we use CropSuite data without climate variability, since the GAEZ approach does not consider climate variability as well. Coffee was compared against the best type of robusta and arabica, as done in the GAEZ data (Fischer et al., 2021).Overall, there are large overlaps between the GAEZ and CropSuite (Fig. 8). Generally, CropSuite identifies larger suitable areas than GAEZ for Africa (purple bar in Fig. 8), particularly for barley, cabbage, chickpea, rapeseed, rye and wheat. A main reason for differences may be due to different underlying soil data, GAEZ uses the HWSD while CropSuite uses the SoilGrids data. As an example, we found abrupt changes in the GAEZ results, especially between borders (e.g. between Angola and Zambia), which follows patterns of the underlying HWSD, which is a known issue (Dewitte et al., 2013). The consideration of climate variability in CropSuite mainly results in larger areas that are unsuitable in CropSuite but still suitable in GAEZv4 (orange bars) (Fig. S4).

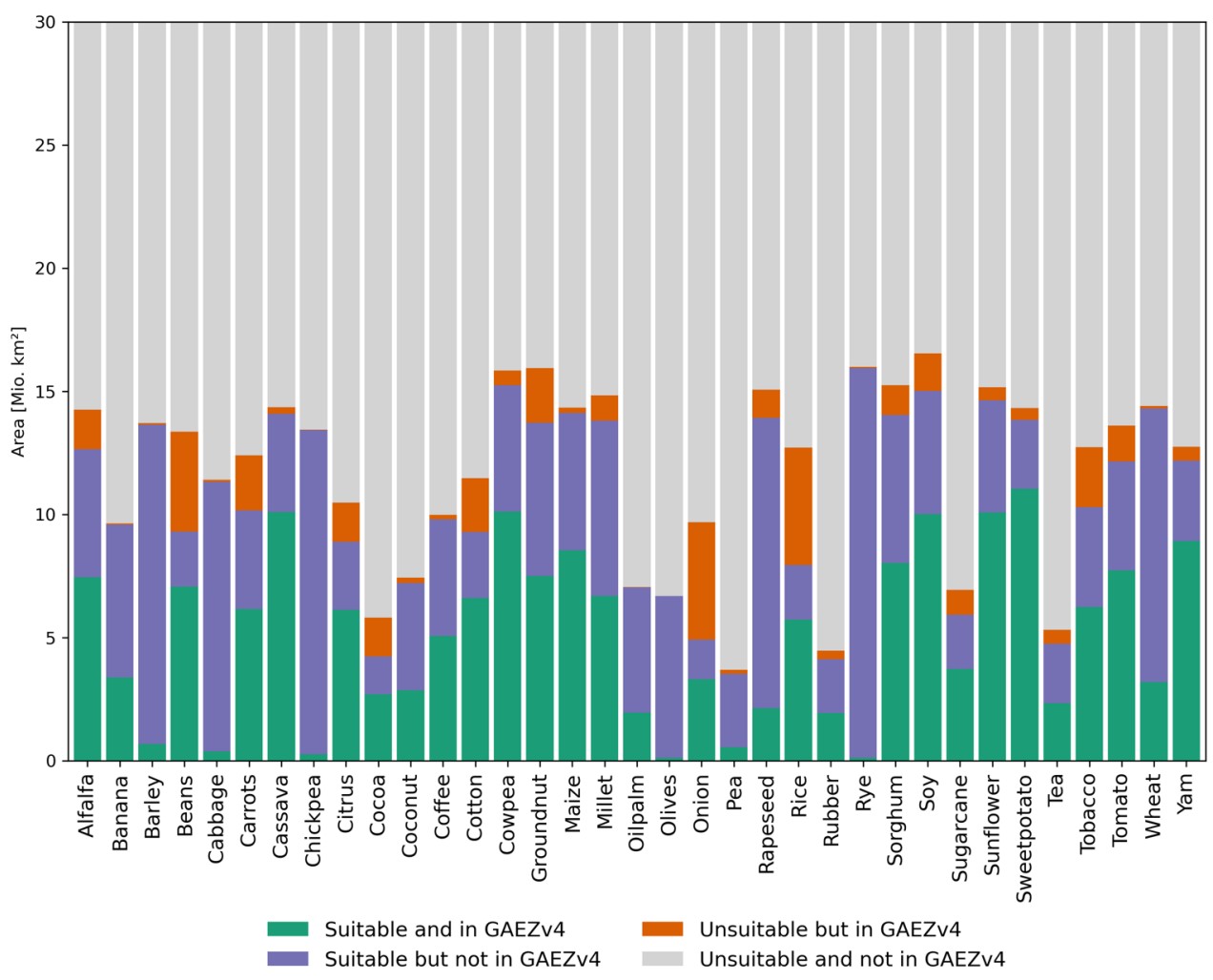

332

**Figure 8: Comparison between CropSuite and GAEZv4 suitability data for all matching crops.** CropSuite results are shown without consideration of climate variability. Areas that are suitable in both data, CropSuite and GAEZv4 are shown in green, areas suitable in CropSuite but not suitable in GAEZv4 are shown in purple. Unsuitable area in CropSuite that is suitable in GAEZv4 is shown in orange. Areas that are unsuitable in both data are shown in gray.

**3.3 Comparison of Optimal Sowing Dates with the GGCMI Crop Calendar**

Another method of validation involves comparing the optimal sowing dates computed with CropSuite with the crop calendar from the Global Gridded Crop Model Intercomparison (GGCMI), which is available globally for a variety of different crops at half degree spatial resolution (Jägermeyr et al., 2021). Figure 9 illustrates the average differences of the sowing dates across Africa, averaged for the matching crops between the two datasets. The comparison is performed at a spatial resolution of 30 arc seconds (Fig. 9) and at half degree resolution (see Fig. S5). For the high spatial resolution, the GGCMI data are interpolated to 30 arc seconds using nearest neighbor. Unlike CropSuite, which displays the optimal

sowing date, the GGCMI data show the actual sowing date based on extrapolated statistics. Thus, there might be differences between the optimal and actual sowing dates. It must also be considered that the GGCMI crop calendar is based on statistics that apply to discrete areas at relatively coarse half degree spatial resolution, while CropSuite was simulated at a pixel accuracy of 30 arc seconds spatial resolution. In fact, the median differences are mostly within one month of the GGCMI crop calendar, which generally indicates a high agreement. Generally, we found that a greater distance to the equator potentially increased the discrepancy between the two data. As an example, in tropical climates with occurring dry and rainy seasons, a shift from one rainy season to another rainy season might result in a greater discrepancy. Also, we found that the distribution of sowing dates over the year was less concentrated in CropSuite, which could be a result of the higher spatial resolution (see Fig. S6). At the coarse resolution, the difference between the two datasets is less and the spread is smaller (Fig. S5).

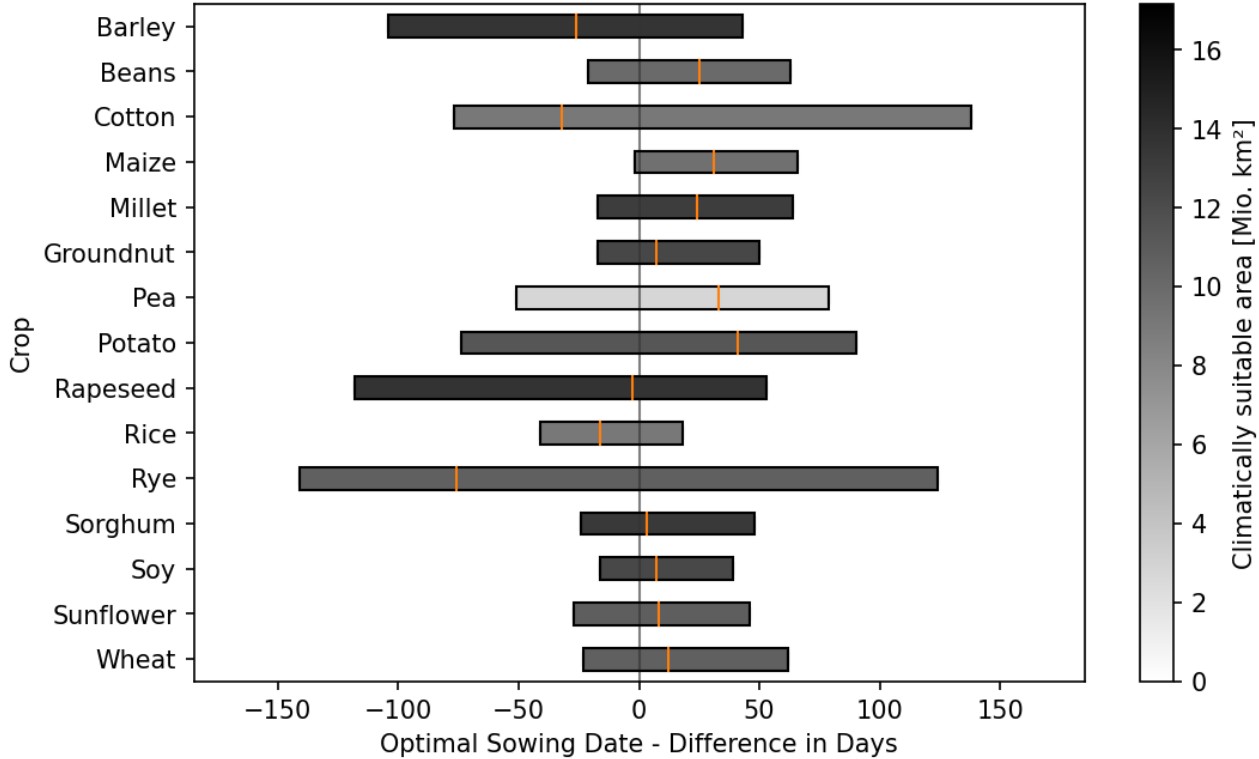

**Figure 9: Comparison of the optimal sowing dates of CropSuite with the actual sowing dates of the GGCMI crop calendars.** The area-weighted shift of the sowing date in days is shown for all matching crops. Negative values mean an earlier sowing date in CropSuite, positive values mean a later sowing date in CropSuite compared to the GGCMI Crop Calendar. The bars show the 5th and 95th percentile, the orange marker shows the median. The color of the bars indicates the climatically suitable area for the whole of Africa. Irrigated areas are considered according to Meier et al. (2018). The comparison is performed at 30 arc seconds spatial resolution for both datasets.

## 4 Simulation Results

Crop suitability is simulated for historical climate conditions (1991-2010) for rainfed and irrigated conditions. Figure 10a illustrates the overall crop suitability, showing for each location the value for the most suitable of all considered crops. Irrigation is considered according to the currently irrigated areas for Africa (Meier et al., 2018), such as along the Nile river in Egypt (see Fig. S1 for irrigated areas in Africa). In total for Africa, 5.7 million km$^2$ are highly suitable, 10.6 million km$^2$ are moderately suitable, 3.3 million km$^2$ are marginally suitable and 10.4 million km$^2$ are not suitable for crop cultivation. Mainly between 10° N and 10° S, a high potential for multiple cropping exists with the possibility of two or three harvests per year (Fig. 10b). Looking at the number of crops suitable for cultivation (Fig. 10c), a large proportion of the considered crops can grow particularly along the wet savannahs, which gives these regions plenty of opportunities for cultivation. In contrast, only a few crops are suitable for the inner tropics and the dry savannahs, which limits the possibilities for switching between crops.

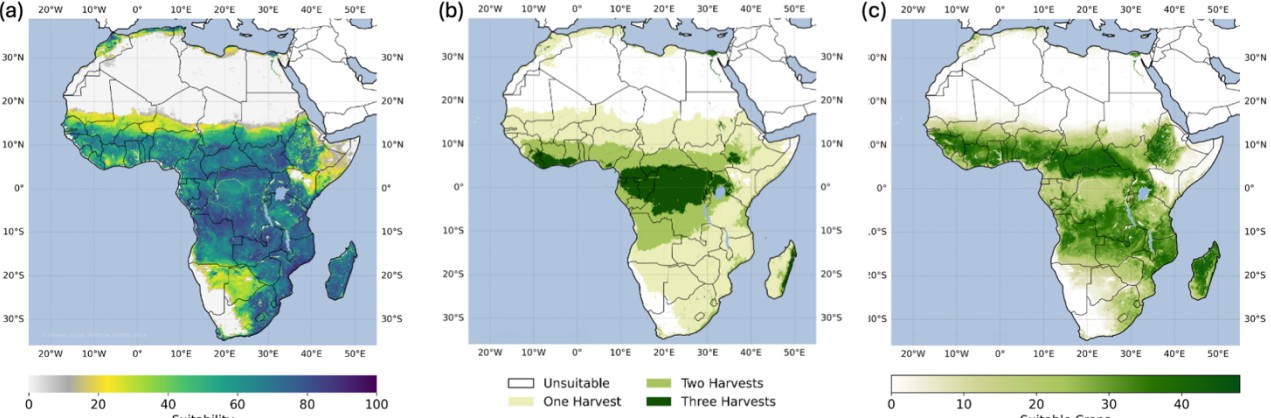

**Figure 10: (a) Overall crop suitability, (b) potential multiple cropping, and (c) number of suitable crops under historical climate conditions from 1991 to 2010.** Irrigated areas are considered according to Meier et al. (2018). The overall crop suitability (a) and the potential multiple cropping (b) are each shown for the most suitable crop at each location. The maximal number of suitable crops results from the number of 48 considered crops (see Table 1). Figure 10a is shown with different colormap in the supplement (Fig. S7).

Figure 11 shows the suitable area for each of the simulated crops for Africa. The five crops with the largest suitable areas in Africa are safflower (16.82 mio km2), sesame (15.76), guava (14.15), cowpea (13.61), and mango (13.39).

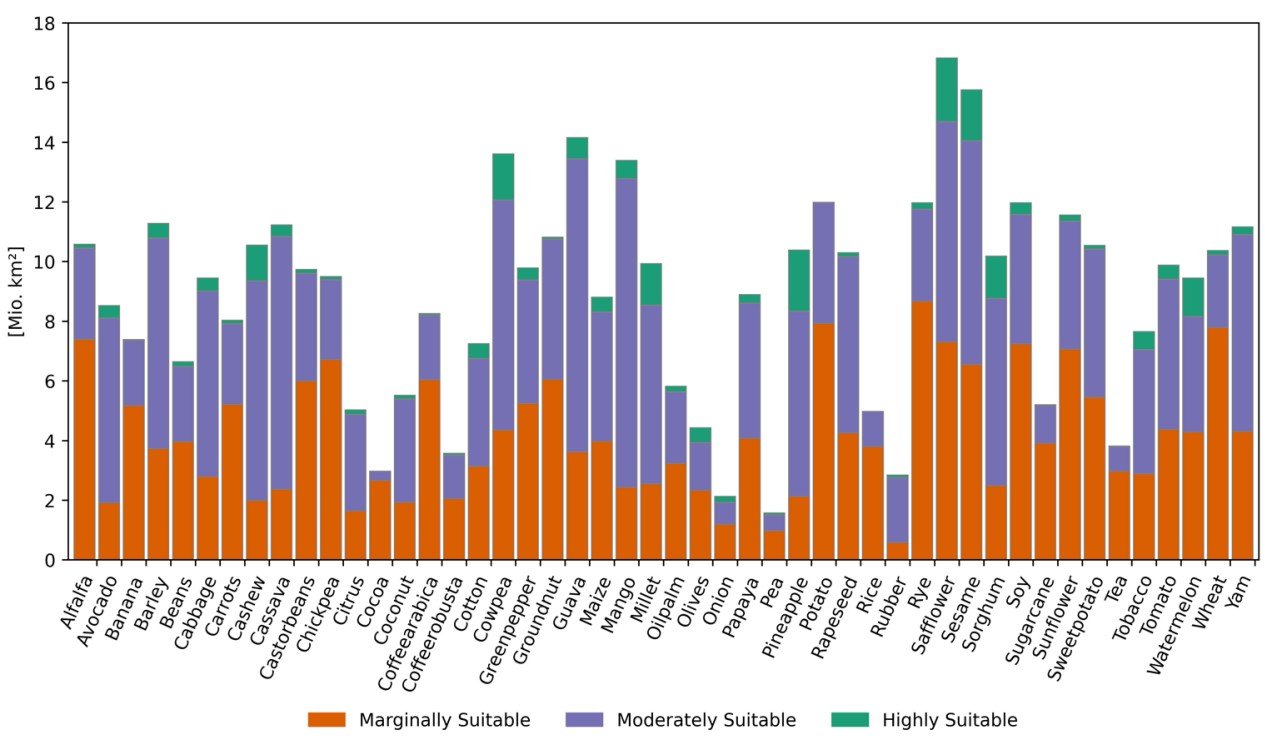

380

**Figure 11: Marginally, moderately and highly suitable areas for all 48 crops under historical climate conditions from 1991 to 2010 for Africa.** Suitability classes are chosen according to Table 3. Irrigated areas are considered according to Meier et al. (2018).

Figure 12a exemplarily shows the crop suitability simulated for maize. The maps for all crops are provided via Zenodo (see Data Availability). Maize is highly suitable along a strip of the $10°$ N and the $20°$ S parallel as well as large parts of Mozambique and Madagascar. In total, 0.49 million km$^2$ are highly suitable, 4.34 million km$^2$ are moderately suitable, 3.97 million km$^2$ are marginally suitable and 21.23 million km$^2$ are unsuitable.

The optimal sowing date for single cropping (Fig. 12b) for maize shifts with latitude from the northern hemisphere across the equator to the southern hemisphere. Figure 12c shows the potential number of potential harvests per year for maize. Climate conditions allow up to two harvests per year in some parts of Congo and Cameroon and in the irrigated areas e.g. along the Nile river. Optimal sowing dates for first and second sowing on areas suitable for multiple cropping are shown in Fig. S8.

Figure 12d shows the climate suitability for maize, which just considers climatic constraints for the suitability of maize. In comparison to the crop suitability map (Fig. 12a), more areas are suitable and suitability is substantially higher, if soil and topography are not considered and therefore do not limit or reduce crop suitability.

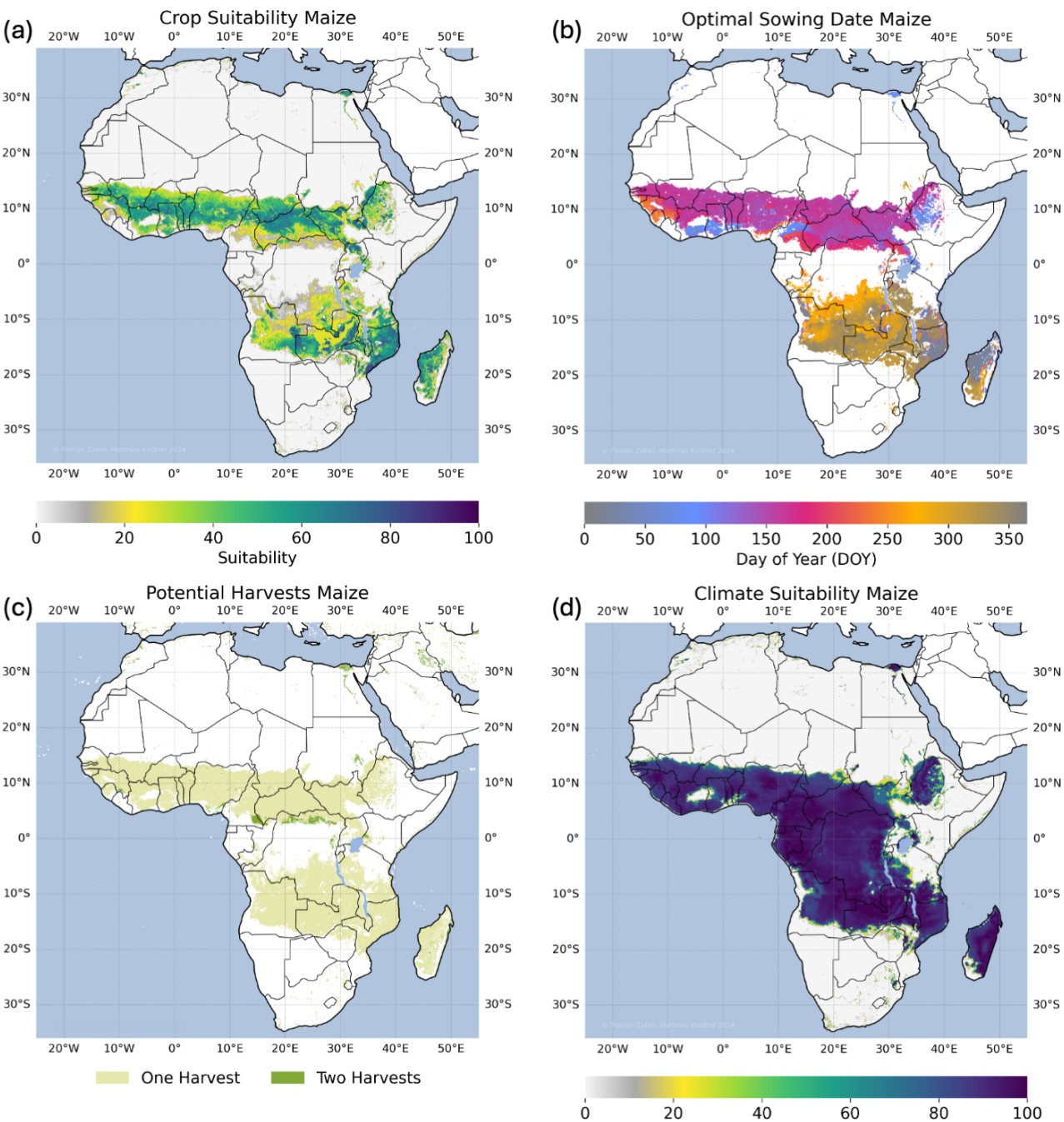

**Figure 12: (a) Crop suitability, (b) optimal sowing date for single cropping, (c) potential multiple cropping, and (d) climate suitability for maize under historical climate conditions from 1991 to 2010.** Irrigated areas are considered according to Meier et al. (2018). Figure 12a is shown with different colormap in the supplement (Fig. S9).

The most limiting factor is shown in Fig. 13a. While low precipitation prevents maize from being suitable in large parts
of Africa in the arid deserts, soil is predominantly restricting suitability in tropical regions. Particularly pH is the most
limiting factor in the humid tropics, such as the Congo Basin, where soils are too acid for growing maize. A large band
along the drylands highlights regions where inter-annual climate variability is most limiting maize suitability (in orange,
Fig. 13a). Here, climate conditions are instable for maize cultivation, and the recurrence rate of potential crop failures is
larger than 25% (every fourth year). For maize, climate variability is limiting crop suitability on 4.4 million $km^2$ for
Africa (Fig 13a).
Figure 13b shows the degree of limitation for all considered climate, soil and terrain factors along a transect following
the 20° E from North to South. In the Sahara, several factors, including temperature, organic carbon content, and soil pH,
are not in an optimal range, while precipitation and the climate variability are the most limiting (note that climate
variability is by definition a limiting factor if precipitation and/or temperature are limiting factors). Due to the unfavorable
soil conditions, irrigation would only slightly improve maize suitability here. Between 15° N and 5° N, the limitations of
all factors are relatively low. Here, coarse fragments and base saturation are most limiting. The tropical areas along the
transect between 5° N and 10° S are mainly constrained by soil pH. Accordingly, soil management or practices that
increase pH in these regions would have a significantly positive impact on crop suitability in this region, since no other
factor has such a strong impact on maize suitability. Further south, low precipitation again mostly limits maize suitability.

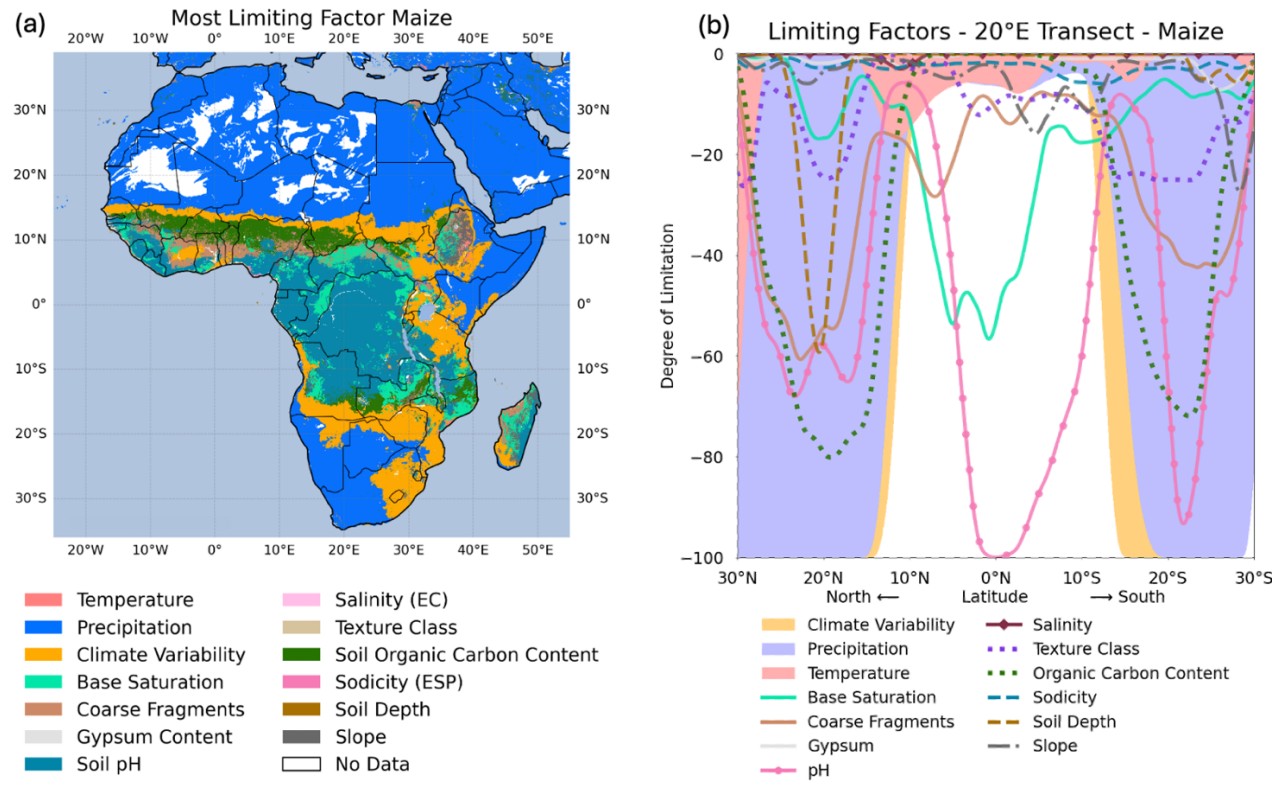


**Figure 13: Limiting factors.** (a) Most limiting factor of the crop suitability for maize under historical climate conditions from 1991 to 2010. (b) shows the degree of limitation of all factors along a transect of the 20° East from 30° North to 30° South. The most limiting factors are displayed with priority according to the order in the legend in (a), if more than one factor fully limits the suitability. For visualization, the shapes in (b) are smoothed using a moving average. Irrigated areas are considered according to Meier et al. (2018) in (a) and are not considered in (b).

The consideration of climate variability significantly reduces climate suitability for maize as shown in Fig. 14a, mainly in the transition area between dry savannah and desert in the Sahel zone, in Burundi and Tanzania in Eastern Africa, and in the southern part of Africa in Angola, Zambia, Zimbabwe, Mozambique, South Africa, and the southern part of Madagascar. In total, climate variability reduces climate suitability on more than 5.4 million km².

Optimal sowing dates also shift when considering climate variability, since the algorithm identifies the best suitable time window for the growing cycle over the year (Fig. S10). As a result, optimal sowing for maize considerably shifts in Tanzania, Mozambique and Madagascar.

Over all crops, Fig. 14b shows the impact of climate variability on the overall crop suitability. In this case, overall crop suitability is reduced on 2.2 million km², mainly reduced in Somalia, Kenya, Ethiopia, South Africa, and the Maghreb countries of Morocco, Algeria, Tunisia, and Libya. These regions generally show a high vulnerability to climatic variability. Climate variability also reduces the potential for multiple cropping in general over all crops on more than 2.3 million km² (Fig. S11).

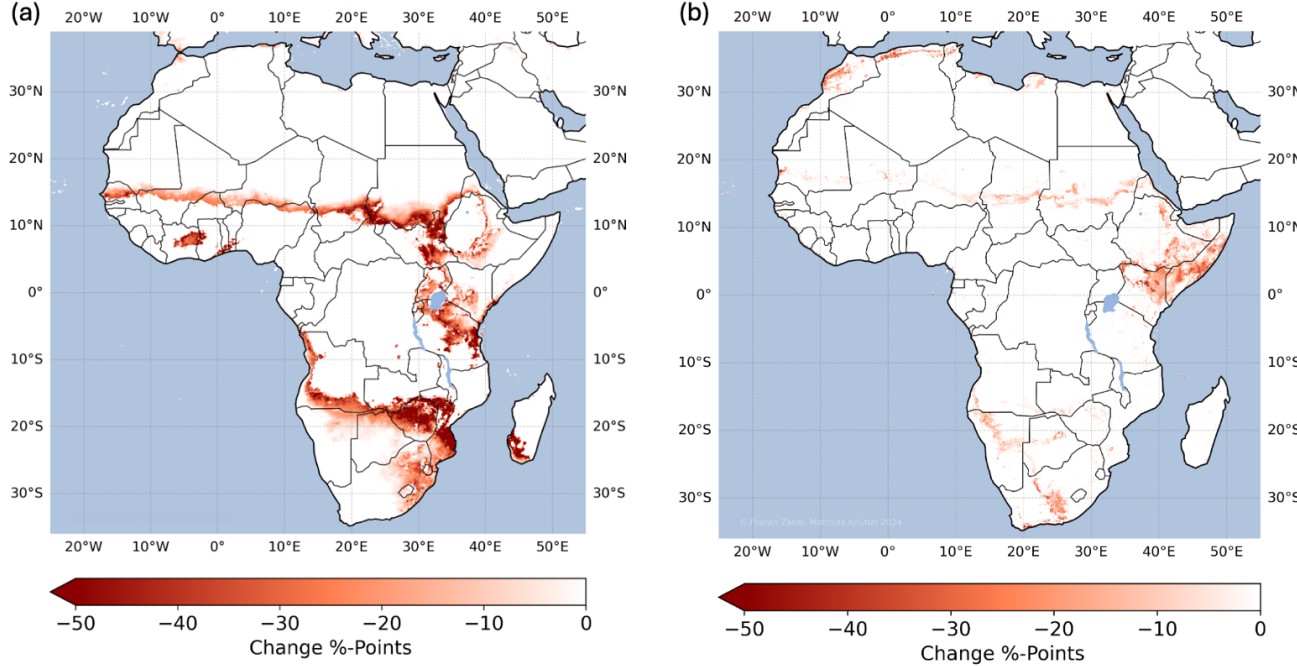

**Figure 14: Impact of the consideration of climate variability on crop suitability (a) for maize (b) for the overall crop suitability of all crops under historical climate conditions from 1991 to 2010.** Irrigated areas are considered according to Meier et al. (2018).

## 5 Discussion

We found that the consideration of climate variability significantly affects crop suitability, multiple cropping, and optimal sowing dates in Africa. Our approach allows to adjust the risk aversion of farmers by adjusting the thresholds for climate variability (section 2.2.) and the membership function (Fig. 5). The shape of this function may differ between crops and regions and might be influenced by several socio-economic factors, such as the degree of mechanization, financial possibilities, and the availability of crop insurances, which is likely to reduce risk aversion of farmers. We suggest the function as shown in Fig. 5 as a broad and general solution which is primarily designed to represent risk aversion of commercial farms. In our comparison analysis for maize (section 3), reference data showed some cultivation in the regions we identified as unsuitable due to the high recurrence rate of potential crop failures caused by high climate variability (Fig. 7). In some regions, despite the high risk of crop failures, land might be cultivated by smallholders or subsistence farmers that have no other choice but to cultivate these lands. However, we admit that the tuning of the climate variability thresholds and the membership function requires more research, and the optimal results will vary depending on crop and region. CropSuite offers the platform and the possibilities to conduct such assessments.

The results of CropSuite (section 4) are subject to uncertainties in the applied climate, soil, terrain, and irrigation data as well as the membership functions (Fig. 1). Soil and terrain data are assumed to be static, although management could influence soil properties such as pH, and terracing could reduce slope limitations. The applied climate data from CHIRPS and CHIRTS are found to be particularly valuable in regions, where climate stations are sparse. Over Africa, CHIRPS is successfully validated (Dinku et al., 2018) showing good performance (Lemma et al., 2019; Muthoni et al., 2019). Verdin et al. (2020) also report good agreement of CHIRTS over Africa, however with a poor performance over central Africa, the Horn of Africa, and parts of northern Mali. Generally, both data sets rely on station data to correct the satellite estimations, which is why uncertainties for very data-scarce regions remain. To apply CropSuite in regions outside 50°S-50°N, or to larger time periods before the 1980s, the user of CropSuite could also rely on global high-resolution climate reanalysis, such as ERA5 (Hersbach et al., 2020). For the African continent, ERA5 reanalysis shows large improvements over its predecessor ERA-Interim (Gleixner et al., 2020). Still, considerable deviations in precipitation from CHIRPS are reported, e.g., wet biases over Uganda (Gleixner et al., 2020) and a dry bias over the western Sahel (Gbode et al., 2023), where CHIRPS is applied as reference. We therefore assume that CHIRPS and CHIRTS are very suitable climatic data sets to investigate our example of maize suitability in Africa. The soil profiles used for the generation of the SoilGrids show a heterogeneous distribution, with large gaps over central Africa, which is why Hengl et al. (2017) attribute uncertainty in the data to the under-sampling. They argue that a few hundred additional profiles in under-sampled areas could massively improve the resulting SoilGrids.

The membership functions derived by Sys et al. (1993) are widely applied but are also governed by inherent uncertainties. Herzberg et al. (2019) argue that the assessment by Sys et al. (1993) is not detailed enough to capture specific features of small areas. They find that Sys et al. (1993) would consider a hilly area in tropical Vietnam unsuitable due to too acidic

soils and steep slopes, whereas the local farmers can cultivate the land. Furthermore, the approach cannot account for compound effects and interactions of the climate and soil variables (Elsheikh et al., 2013). The membership functions cover the general behavior in a univariate manner, while the real plant physiology is a more complex interplay of climatic variables and soil conditions (Joswig et al., 2022). This also applies particularly to compound extremes, for example the combination of hot and dry climatic conditions (Goulart et al., 2023) that limit water availability and favor evaporation, which can trigger water and temperature stress in plants. This is relevant in the course of a warming climate, as the joint probability of hot and dry conditions is projected to increase in many regions of the world (Bevacqua et al., 2022; Felsche et al., 2024). This is however no specific drawback of CropSuite, but rather a lack of bivariate, multivariate or interactive membership functions. The assessment of the membership functions by Sys et al. (1993) is also outdated for new crop varieties that might be more resilient to climatic and environmental stressors (Peter et al., 2020). Furthermore, we argue that the uncertainty in the temperature and precipitation membership functions is by design larger at its low and high ends, as the functions are derived empirically. Since our consideration of climate variability is based on the 5% to 10% suitability values, respectively (see Section 2.2), the uncertainties of the membership functions are propagated to the assessment of climate variability. More research and updated functions could support the results by CropSuite.

The sampling of climate variability within 20-year periods is limited as variability can cover wide time ranges. There, the application of single-model initial condition large ensembles can help to robustly assess the variability based on decadal or multidecadal time periods (Deser et al., 2020). This is especially important for precipitation and precipitation extremes, which show a high sensitivity to climate variability (Lang and Poschlod, 2024; Tebaldi et al., 2021). Furthermore, for the assessment of climate variability, we only capture the occurrence of growing seasons exceeding the percentile thresholds, but we do not consider the intensity of the according events. Single days with extreme precipitation can induce flooding that leads to crop failures (Balgah et al., 2023; Müller et al., 2023), even though the average precipitation for the growing season is still within the suitable range of the membership function. This drawback however also applies for most of the mechanistic crop models at global scale (Ruane et al., 2017), while regional applications evolve incorporating crop losses due to waterlogging and flooding (Li et al., 2016; Monteleone et al., 2023; Pasley et al., 2020). This is why we claim to assess climate variability not climate extremes inducing potential crop failures.

**6 Conclusions**

CropSuite is a new easy-to-use comprehensive open-source model that provides a complete processing chain (preprocessing, spatial downscaling, suitability simulations, data analysis and visualization) for carrying out crop suitability and climate change impact analysis. CropSuite allows users to easily parameterize different varieties of the same crops or additional crops by determining the membership functions in the GUI. Thereby, the fuzzy logic approach makes it easy to use expert knowledge for the parameterization of the membership functions. Besides all data and compiled maps generated, we provide a user manual for CropSuite (Zabel and Knüttel, 2024) and the parameterizations

of the considered 48 crops in this study. Furthermore, the model allows the flexible addition of further parameters and membership functions that might affect suitability, if the required data is provided. For the future, this allows the consideration of further ecological and socio-economic limitations (such as access to fertilizers, available labor, know-how, infrastructure and transportation, heat stress impacts on labor) that have not yet been sufficiently considered in crop suitability assessments (Orlov et al., 2024; Akpoti et al., 2019).

For this study, we simulated 48 crops for Africa under the consideration of climate variability for historical climate conditions. Thus, we created a huge dataset, providing detailed high-resolution information on climate-, soil-, and crop suitability, optimal sowing dates, multiple cropping potentials and the limiting factors, which can be used for follow-up studies and climate impact assessments. Additionally, the data include substantial information to develop strategies for an efficient land-use (Schneider et al., 2024; Molina Bacca et al., 2023; Delzeit et al., 2019). The consideration of future climate change scenarios will allow for investigating efficient strategies for climate change adaptation through shifting sowing dates, or cultivar and land-use change. Further, information about the limiting factors can be helpful to optimize crop management, since it identifies the parameter that most efficiently improves crop suitability.

**Code Availability**

CropSuite (v1.0) code is written in Python and is available Open-Source (CC BY-SA 4.0) together with the GUI at Zenodo (https://doi.org/10.5281/zenodo.14259375) and GitHub (https://github.com/flozabel/CropSuite). A user manual is provided separately via Zenodo (https://doi.org/10.5281/zenodo.14196315).

**Data Availability**

The resulting data are available for download as GeoTIFF files via Zenodo (https://doi.org/10.5281/zenodo.14514729). In addition to the figures shown as examples for maize in this paper, the compiled figures for all 48 considered crops are provided for download, including a separation of rainfed and irrigated agricultural systems and a comparison with MapSPAM 2020 (https://doi.org/10.5281/zenodo.14514729).

**Author contribution**

FZ conceptualized and developed the model. MK programmed the CropSuite model and the GUI in Python. FZ, MK, and BP developed the methodology for the consideration of climate variability. FZ and MK performed the simulations and analyzed the results. FZ and MK prepared the manuscript with contributions from BP.

**Competing interests**
The authors declare that they have no conflict of interest.
**Acknowledgements**
The simulations were performed at sciCORE (http://scicore.unibas.ch/) scientific computing center at University of
Basel, requiring in total approximately 150.000 CPUh. We thank CGIAR and CIAT for their support and the scholarship
provided to MK and the collaboration for the Africa Agriculture Adaptation Atlas.

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
