# Peer review of "CropSuite v1.0 - A comprehensive open-source crop"

_EGUsphere, 2024_

## Author Comment (AC1)

**Reviewer #1:**

General comments
In this manuscript, Zabel et al. describe a new piece of software, CropSuite, that generates maps of crop suitability and related information based on climate, soils, and terrain. This builds on previous work by themselves and other authors to include, importantly, (a) a consideration of climate *variability* in addition to averages and (b) less-widespread but regionally important crop types. Noting that such crops are under-studied but are especially important in Africa, the authors focus their analyses there. The results look reasonable when compared to real-world crop distributions and sowing dates.

One of the goals of CropSuite was to make something that is easy-to-use and flexible enough to be used by a variety of stakeholders, not just scientists. As a scientist, I can't really assess how accessible it is to less-technical users, but the inclusion of a graphical user interface (GUI) is a really important development. I do think, however, that this tool will be useful to scientists and model developers as well. Global gridded crop models and especially integrated assessment models need to be able to endogenously represent things like sowing date, the potential for multiple cropping, and shifts in what crops are planted where; tools like CropSuite can help.

That said, I do have some questions and concerns about the manuscript as currently written. Thus, I recommend it be considered for publication after minor revisions. See attachment for details.

**Reply:** Dear Reviewer #1, thank you very much for your time to review our paper! We appreciate your thoughtful comments and suggestions a lot! These were really helpful and we were able to improve our study accordingly.
Since the initial submission of this paper, we were able to improve the CropSuite model and the GUI. We uploaded an updated version (v1.0) of CropSuite to Zenodo and GitHub. In addition, we uploaded the complete GeoTIFF dataset and the compiled maps for all 48 crops to Zenodo.
We also hope that CropSuite will be further used not only by stakeholders, but also be further developed by scientists and model developers, which is our main motivation to provide the source code of CropSuite as open source. We prominently added this goal to the end of the abstract, since this was hidden so far. We also agree that the development of the GUI is an important aspect that may have been somewhat neglected in our paper. To highlight the importance of the GUI, we added it also to the abstract.
Thank you very much that you recognize the potential of CropSuite to improve crop models and integrated assessment models, which is very motivating for us.

In the following, we refer to your comments and answer them directly below your comment. Please note that line numbers in our reply refer to the revised version of the manuscript with track changes. Thank you!

Specific comments
• L53: It would be really helpful to have a single place where you briefly list all the new things in this version (other than the GUI). Is it just the addition of climate variability, new crops, and new pre-/post-processing tools? Or are there other things?

**Reply:** At the end of the user manual, which is now provided as PDF via an individual Zenodo link (https://doi.org/10.5281/zenodo.14196315), we provide a 4-page changelog in Chapter 8 that describes in detail all changes in the model. In the paper, we concentrate only on the main conceptional changes of the model, since we cannot describe all minor improvements that are more technical issues. Other than the GUI, this is mainly the consideration of climate variability and the addition of pre- and postprocessing tools, which are included to allow technically less experienced users to be able to apply the model and analyze the results. The simulation of additional crops is rather a parameterization issue than a model development.
To mention the existence of a user manual, we added a line at the end of the introduction and refer to it.

• L104: Why change the classification system from 6 to 4? Without a clearly-elaborated reason, a cynical reader might think this was done to make comparisons more favorable.

**Reply:** The reason for changing the classification system from 6 to 4 classes in our study was to simplify the categories. Therefore, we merged the classes N2 (unsuitable) and N1 (actually unsuitable and potentially suitable), also because of a very unspecific and vague definition of N1. We interpretate N1 as unsuitable. As a result of the reclassification, our results show a wider contrast of suitable land, which might be beneficial e.g. for comparison between crops. In principle, the classification system is not hard-coded in CropSuite and can be set by the user. It is important to ensure that the membership functions match the selected classification.

• How is irrigated area considered in CropSuite? Is there some input dataset about the area equipped for irrigation? I ask because I had thought CropSuite would just run calculations and produce figures for rainfed and irrigated crops separately, but Fig. 7 doesn't say whether it's for rainfed or irrigated datasets, and the associated text suggests discrepancies are due to "different assumptions on irrigated areas. " (Later— first in the caption of Fig. 9—I see that "Irrigated areas are considered according to Meier et al. (2018). " Is that the case for all analyses? This should be explained in the Methods.)

**Reply:** Irrigation is an option that can be switched off and on (also via the GUI). We simulated the considered crops for both, irrigated and rainfed options. In the post-processing, we combined both datasets according to the equipped area for irrigation (Maier et al. 2018), which is available at 30 arc seconds resolution (see Fig. S1).
This is indeed not explained in the Methods so far and we therefore added a paragraph in line 196-199.

• Comparison to GGCMI sowing dates
o Dates must not be bilinearly interpolated unless the algorithm is smart enough to account for the modulo nature of dates. For example, interpolating between Jan. 3 (day 3) and Dec. 30 (day 364) should give Jan. 1 (day 1), but a not-modulo-capable linear interpolation would give July 4 (day [3+364]/2 = 183.5). Instead of bilinear interpolation to downscale the GGCMI dates, please switch to nearest-neighbor (or explain what tool you're using that can handle modulo interpolation—I'd love to have one!).

**Reply:** Thank you for pointing this out! We recalculated Fig. 9 using nearest neighbor (see below). See also next comment.

[Figure]

Figure 9: Comparison of the optimal sowing dates of CropSuite with the actual sowing dates of the GGCMI crop calendars. The area-weighted shift of the sowing date in days is shown for all matching crops. Negative values mean an earlier sowing date in CropSuite, positive values mean a later sowing date in CropSuite compared to the GGCMI Crop Calendar. The bars show the 5th and 95th percentile, the orange marker shows the median. The color of the bars indicates the climatically suitable area for the whole of Africa. Irrigated areas are considered according to Meier et al. (2018). The comparison is performed at 30 arc seconds spatial resolution for both datasets.

o Alternatively, you could coarsen the CropSuite outputs to match the GGCMI resolution. This would allow you to avoid the interpolation issue: Go one-by-one through the GGCMI cells, modulo-summing all the CropSuite dates, then divide by the number of CropSuite cells in a GGCMI cell. It would also avoid the issue you describe at lines 286–288, of CropSuite simulating a much higher spatial resolution than GGCMI provides.

**Reply:** We included the comparison at the half degree resolution in Fig. S5 and therefore resampled the CropSuite optimal sowing dates from 30 arc seconds to half degree resolution by using the modulo (ignoring NA values). At the coarse resolution, the difference between the two datasets is less and the spread is slightly smaller, as expected. We changed the description(L344-349).

[Figure]

Figure S5: Same as Fig. 9, but the comparison is performed at half degree spatial resolution for both, the GGCMI and the CropSuite crop calendar datasets. Therefore, the CropSuite data is resampled from 30 arc seconds to half degree spatial resolution using the modulo (most frequent value within a corresponding coarse pixel).

o Note also that, according to the supplement of Jägermeyr et al. (2021), the GGCMI crop calendar product contains no interpolation (as you state, "the GGCMI data show the actual sowing date based on interpolated statistics"), but only extrapolation.

**Reply:** Good point, thank you! It is based on extrapolated statistics. We corrected that!

o Do there tend to be any patterns in the discrepancies that might explain them? E.g., do the poorly-matching crops tend to have longer growing seasons? Do the discrepancies tend to happen more in certain parts of the world?

**Reply:** We tried to identify systematic patterns and created histograms and maps for all crops. One pattern we found is that the greater the distance to the equator, the greater the discrepancy was in general. This can be the case in tropical climates with occuring dry and rainy seasons, where a shift from one rainy season to another rainy season might result in a greater discrepancy. Also, we found that the distribution in the histogram was less concentrated, which is probably a result of the higher spatial resolution of CropSuite. The following figure shows the histograms of the sowing dates for Africa of CropSuite in comparison with the GGCMI crop calendar for Sorghum as an example.

[Figure]

Optimal Sowing Data Sorghum

GGCMI Crop Calendar    CropSuite

• Various figures: Please avoid red and green on the same plot, as that's hard to read for people with the most common form of color vision deficiency. This is less of an issue on some plots than others, with the maps being especially bad and Fig. 10a probably the worst. You can use the Color Blindness Simulator at https://www.color-blindness.com/coblis-color-blindness-simulator/ to check accessibility of color schemes; ColorBrewer has some good suggestions at https://colorbrewer2.org/#type=qualitative&scheme=Dark2&n=3 (make sure to click "colorblind safe). The "Choosing colormaps in Matplotlib" webpage (https://matplotlib.org/stable/users/explain/colors/colormaps.html) also has a lot of good guidance.

**Reply:** Thank you for nudging us using the ColorBrewer! We changed our colors accordingly to make it color blind friendly. Therefore, we revised Fig. 6, Fig. 7, Fig. 8, Fig. 11. Regarding Fig. 10a, 12a, 12b, and 12d, we additionally provide maps with a colorblind friendly colormap via Zenodo for all crops (file name '_colorblind'). For these figures, we would like to stick with the current colormap, because this is a widely used colormap for illustrating suitability, which allows for easier comparison. GAEZ for example uses a very similar colormap for mapping their results (https://gaez-services.fao.org/apps/theme-4/).

o For Figs. 10(a) and 12(a, d), consider using a qualitative colormap with your three suitability categories instead of a gradient. As it is now, the abrupt changes in the gradient don't really correspond to the boundaries between your categories, and thus it's hard to compare to the text at L300–301. Moreover, it's totally impossible to read with red- or green-blindness.

**Reply:** We thought about using classes instead of a gradient, but finally decided against, since the categories would remove a lot of interesting information and heterogeneity in the figure. The categories used in Fig. 11 are described in Table 3. We are not sure what you mean with abrupt changes in the gradient, because the color bar doesn't have abrupt changes and is quite gradual (if that is what you mean). Maybe, this results from the bad resolution of the figures in the preprint. The original figures don't show that.

Minor comments
• For GMD, title must include version number.
**Reply:** Thanks for the hint. For the review, we used CropSuite version 0.9. For the final paper, we updated the source code and data on Zenodo and GitHub to version 1.0. We also included the version number to the title.
• L51: Briefly explain what "fuzzy logic" is.
**Reply:** Fuzzy logic is a music album of the Super Furry Animals from the year 1996 (highly recommended!). In our context, it is a form of logic theory. In contrast to Boolean logic, truth value of variables may be any real number between 0 and 1 in fuzzy logic. We added a sentence generally describing the idea of fuzzy logic in L59.
• L52: Why are they called "membership functions"?
**Reply:** In fuzzy logic, fuzzy sets consist of elements that have degrees of memberships that are described in membership functions (Zadeh L.A., 1965). [Fuzzy logic doesn't describe likelihood or probability but rather represents a membership in vaguely defined sets]. We added a general explanation to membership functions in L60.
• Table 1: Errors in Latin names—spaces in wrong place, "ti" glyphs missing, italics missing, some other misspellings. E.g., "Chickpea (Cicerorie num)" should be "Chickpea (Cicer arietinum)"; "Rice (Oryza sa va)" should be "Rice (Oryza sativa)".
**Reply:** Thanks a lot for looking so closely! Something went wrong here when copying the table to the document, also the order was disrupted. Sorry that we haven't noticed that before. We replaced the whole table made the latin name italic and double checked everything.
• Table 2:
o Include citations
**Reply:** Done.
o Sodicity row missing "ISRIC Harmonized Dataset of Derived Soil"
**Reply:** We added that!
• Fig. 1, temperature: Is that mean temperature over the 110 days? Based on later text it looks like yes; maybe this could be mentioned on the figure or in its caption.
**Reply:** Yes, that's correct! The length of the growing cycle is already mentioned in the figure caption. We added mean temperature over the growing cycle and total precipitation over the growing cycle to the figure caption, since it is too long for the figure title.
• L155–156: What's the difference between sowing and planting?
**Reply:** Maize and wheat, for example, are sown, while rice is planted as a seedling in wet rice cultivation.
• Fig. 3: The "climate suitability" plot is hard to read. The "limiting factor" arrow is confusing and should be replaced with a legend instead. Then you could add a dashed black line that traces out the suitability throughout the year, according to which of the three curves is lowest on any day. You basically already have that with the green shading, which I didn't notice before. Again, a legend would have been helpful. And

maybe change it from green (which also represents the climate variability curve) to something not already used (e.g., gray).

**Reply:** Thanks a lot for your suggestions, these totally make sense! We changed the figure accordingly.

• Fig. 5: X axis tick labels should be 0, 10, and 20, not 0, 0.1, and 0.2.

**Reply:** Absolutely, we corrected this!

• Fig. 6: It would be useful to have a version of this figure in the Supplement that had "proportion of MapSPAM area" on the Y-axis. Would probably want to swap the top and bottom colors for this.

**Reply:** We agree and added Fig. S3 to the Supplement! We left a revised version of Figure 6 in the main paper, because the potentially suitable area that are currently not cultivated with the respective crop according to MapSPAM 2020 can additionally be seen here.

[Figure]

Figure S3: Same as Fig. 6 but showing the proportion of MapSPAM area on the y-axis. Green areas are suitable and in MapSPAM 2020, orange areas are unsuitable but in MapSPAM 2020.

• L233: Why MapSPAM 2020 instead of SPAM 2000/2005/2010? Maybe the latter don't include as many of your crops? But could still be useful.

**Reply:** We did this analysis for different versions of MapSPAM, however, for the paper we decided to show only the comparison with MapSPAM 2020, because that was the latest release with a special focus also on Africa. It also provided the largest overlap of crops with our study. MapSPAM 2020 has 32 crops, while SPAM2017 has 29 crops and SPAM2000 only 15 crops. As you can see below in the screenshot, exemplarily for maize, the SPAM data differ a lot. While SPAM2000 shows large harvested areas that are not suitable in CropSuite (red areas), these areas are less in SPAM2017 but considerably

lower in SPAM2020. We included this comparison to Figure S2 and added a sentence in L292-293.

[Figure]

*Figure S2: As Figure 7b but comparison of CropSuite with (a) SPAM2000, (b) SPAM2017 , and (c) MapSPAM 2020, exemplarily for maize.*

• L261: "inner tropics"? What about at the southern edge of the Sahara?

**Reply:** The southern edge of the Sahara is unsuitable in CropSuite because climate variability is limiting here (compare to Figure 7b where climate variability is not considered). This is described in L303-311.

• Fig. 8:

o Why does this plot have gray for "unsuitable and not in GAEZv4" while Fig. 7 had no such color? Also, why is it capped at 30 Mkm2?

**Reply:** We recompiled Fig. 6 and Fig. 8. Now, both also show the class "unsuitable and not in ...". We also rearranged the bars in Fig. 8 according to Fig. 6. The area is capped at 30 Mkm2, because this is the total area of Africa.

o Does GAEZv4 really not have rice?

**Reply:** You are right, rice was overseen, also carrot, coffee, millet (pearl millet), olive, rubber. Coffee was compared against the best type of robusta and arabica, as also done in the GAEZ data. Also, we recognized that the available GAEZ data are only rainfed and cannot be compared to the combined rainfed and irrigated CropSuite data. We corrected that. Accordingly, we revised the comparison with GAEZ and included a clearer description of the GAEZ data used and included the missing crops in our comparison. GAEZ do not consider climate variability which affects the comparison (see also comment reviewer #2). Therefore, we now show the comparison without climate variability in the main paper and included the same figure with consideration of climate variability to the supplement.

• I was pretty surprised after finishing Sect. 3 to see that Sect. 4 was labeled "Results. " What does Sect. 3 do if not present results? Please merge these into one Results section and use sub-headings to explain what each does.

**Reply:** We changed the heading of Sect. 3 to "Model Evaluation" and Sect. 4 to "Simulation Results"

• L306: "cultures" is probably not the right word.

**Reply:** Thank you, we changed cultures to crops

• L312 and Fig. 11: Specify in text and caption that Fig. 11 is for Africa only.

**Reply:** We included that in text and caption.

• Fig. 12c: "Three harvests" color doesn't seem to appear anywhere and thus should be removed for clarity. Color contrast between one and two harvests is very low, making this map hard to read. Higher image resolution would also help readability.

**Reply:** We deleted 'three harvests' from the legend in Fig. 12. Regarding the color contrast between one and two harvests, we agree and changed the colors for Fig. 10b, Fig. 12c and all additional crop-specific maps and figures, provided via Zenodo.
The original figure resolution we provided is much higher, maybe this comes from pdf conversion of the preprint?

• L325–327: use of "where" in this sentence is confusing. Suggested rephrasing: "In comparison to the full crop suitability map (Fig. 12a), more areas are area is suitable and suitability is substantially higher, where if soil and topography do not limit or reduce crop suitability are not considered."

**Reply:** Agree, we changed that to: "In comparison to the crop suitability map (Fig. 12a), more areas are suitable and suitability is substantially higher, if soil and topography are not considered and therefore do not limit or reduce crop suitability."

• Fig. 13 and related text:

o Why not consider irrigated areas in (b)?

**Reply:** I think there are no or just very few irrigated areas along this transect. Irrigated areas would create artefacts along the gradient through the different climate zones. Nevertheless, the shapes in Fig. 13b are smoothed by using moving averages, which would again level out these artefacts. We included the smoothing to the figure caption.

o L339–340: "note that climate variability is by definition a limiting factor if precipitation and/or temperature are limiting factors." However, it seems like—when both are 100% limiting—you've chosen temperature or precipitation for Fig. 13a, not variability. This seems like a good choice but is never explicitly stated. Also, this implies that Crop Failure Frequency in Fig. 13b should be plotted underneath the colors for Temperature and Precipitation—it took me a while to realize that the color at e.g. X[30°N to 20°N] x Y[–100 to –80] was orange on top of blue.

**Reply:** Thank you for pointing this out. We included in the figure caption the following sentence: "The most limiting factors are displayed with priority according to the order in the legend in (a), if more than one factor fully limits the suitability. For visualization, the shapes in (b) are smoothed using a moving average".
According to Fig. 13b, we completely redesigned it for better readability. We think that this figure is valuable, because it is possible to see all limiting factors and their degree of limitation.

o Speaking of colors, it's unfortunate that two of the most important limiting factors across the transect in 13b—precipitation and soil pH—have such similar colors. Consider swapping the latter's for something else.

**Reply:** We completely redesigned Fig. 13b and use area shapes for climate factors and different lines for soil and topography factors, which significantly improves the readability of the figure.

• L365–376:

o Refer to relevant sections and figures throughout.

**Reply:** Done.

o It took me a while to understand the sentence at L370–372. Suggested rephrasing: "In our comparison analysis for maize, reference data showed some cultivation in the regions we identified as unsuitable due to the high recurrence rate of potential crop failures caused by high climate variability (Fig. 7)."

**Reply:** Thank you for your suggestion, we changed that accordingly.

o L374: "Though" should be "However"

**Reply:** We changed that, thank you!

• L378: Delete comma at end of this line to make sentence easier to understand.

**Reply:** Ok.

• L379–380: What sorts of biases?

**Reply:** ERA5 shows precipitation biases (annual precipitation): Gleixner et al. (2020) report a considerable wet bias over Uganda. On the other hand, Gbode et al. (2023: https://doi.org/10.1007/s00376-022-2161-8 ) report a dry bias over the western Sahel. However, for the climate data, we have to correct ourselves as part of the revision. The observational data sets CHIRTS & CHIRPS (based on satellite remote sensing corrected with station observations) were used as climatic input data for the presented CropSuite simulations (and not ERA5). The climate data were provided to us by CGIAR as part of the cooperation for the African Agriculture Adaptation Atlas. We apologize for this confusion. The two datasets are described in the revised version of the article and their advantages and disadvantages are discussed. ERA5 (as a potential driving climate for many CropSuite users) is also briefly discussed.

• L383: Start a new paragraph between these sentences to separate discussion of "input data issues" from "fundamental appropriateness of this method."

**Reply:** Done. Good idea and we like the invisible subheader "fundamental appropriateness of this method".

• L394–395: This sentence is too vague; I don't understand what it means, or why it's set in the middle of a discussion about membership functions.

**Reply:** With this sentence, we want to address another uncertainty of the membership functions, which were assessed by Sys et al. (1993). Since the membership functions are already more than 30 years old, they describe crop requirements of varieties that are 30 years old and do not reflect new crop varieties that might be more resilient or adapted. We rephrased this sentence to be better understandable.

• L395–397: It's unclear what the first part of this sentence ("Furthermore, … membership function") has to do with the second ("which a]ect our consideration of climate variability.

**Reply:** Our consideration of climate variability builds on the values of the temperature and precipitation membership functions at the 5% (or 10%) suitability. These outer ranges of the membership functions are more uncertain than the central parts of the distribution, as these functions are empirically derived. Hence, we argue that our consideration of climate variability suffers from the uncertainties of the membership functions at its outer limits. We clarified this in the article.

• L405: Surely some mechanistic crop models consider flooding.

**Reply:** We clarified our statement: "This drawback however also applies for most of the mechanistic crop models at global scale (Ruane et al., 2017), while regional applications evolve incorporating crop losses due to waterlogging and flooding (Li et al., 2016; Monteleone et al., 2023; Pasley et al., 2020)."

• Comparison_mapspam2020.zip:

o "Unuitable" typo in most files under with_climate_variability/.

**Reply:** Oh, thank you, we fixed in the new uploaded version!

o Why do many files in without_climate_variability/ compare against SPAM2017?

**Reply:** That was a mistake from our side (as mentioned, we performed comparisons with different versions of MapSPAM and forgot to update the legend). Thank you for recognizing that!

• Consider adding to the User's Guide: advice about which climate interpolation function users should choose.

**Reply:** The GUI makes such kind of suggestions by pre-selecting standard methods that already shown up in light grey color before making the selection.

---

## Author Comment (AC2)

**Reviewer #2:**

This study improves the crop suitability assessment method by considering the climate variability. It compares results with SPAM2020, GAEZv4, and the crop calendar from GGCMI. The manuscript is clearly written. I have a few comments.

**Reply:** Dear Reviewer #2, thanks a lot for your comments and for reviewing our paper! In the following, we refer to your comments and answer them directly below your comment. Please note that line numbers in our reply refer to the revised version of the manuscript with track changes. Thank you!

Major:
1. Make the dataset and results more accessible. Now, all the results in Zenodo are *.png, which contain no geographical information. I highly recommend authors to provide geographic file format, e.g., Geotif and netcdf.
**Reply:** It is planned to include the geographic data (GeoTIFF and/or NetCDF) to the Africa Agriculture Adaptation Atlas, where it will be provided for download. However, this will take some time and we don't know when this will happen. Therefore, we uploaded all raw data to Zenodo (https://doi.org/10.5281/zenodo.14196331), where we additionally also provide the compiled maps, that we only show exemplarily for maize in the paper, for all 48 crops as png files via Zenodo. This is not a substitute, but an addition.

 Minor:
1. line 77, Is the soil texture for >200cm also needed? if yes, where does it come from?
**Reply:** Soil texture is calculated based on the clay and sand content taken from SoilGrids data. This data is only available up to 200 cm. Deeper texture is not required.
2. Why the soil layer in lines 79-80 is different with Table 2
**Reply:** This is indeed not well explained and we revised the paragraph and added a description in L96-L101. The paragraph now reads as follows:

Available soil layers can be weighted in CropSuite as required. The SoilGrids datasets provide information for six depths: 0-5 cm, 5-15 cm, 15-30 cm, 30-60 cm, 60-100 cm, and 100-200 cm (Hengl et al., 2017; Hengl et al., 2014). According to Sys et al. (1991), soil properties have different effects on crop suitability depending on the soil layer. Accordingly, we use weighting factors as proposed by Sys et al. (1991) (see Table 2). The different distribution of the soil depths between the SoilGrids data and the weighting factors by Sys et al. (1991) is taken into account by using a proportional weighting of the SoilGrids layers.

3. Line 81, reference formate. And, why are the weights needed? How are these weights applied? It's a bit confusing here. Did the authors mean that weights were used to multiply with original value to generate the new value?
**Reply:** Thank you, we corrected the reference format.
The question is how e.g coarse fragments impact on crop suitability. According to Sys et al. (1993), the upper soil layer (0-25 cm) has a much higher impact on crop suitability than the lowest soil layer (125-150 cm). Therefore, the different available soil layers can be weighted.

4. line 233. OK, MapSPAM2020 may introduce some uncertainties, then why not using MapSPAM2010?

**Reply:** We did this analysis for different versions of MapSPAM, however, for the paper we decided to show only the comparison with MapSPAM 2020, because that was the latest release with a special focus also on Africa. It also provided the largest overlap of crops with our study. MapSPAM 2020 has 32 crops, while SPAM2017 has 29 crops and SPAM2000 only 15 crops. As you can see below in the screenshot, exemplarily for maize, the SPAM data differ a lot. While SPAM2000 shows large harvested areas that are not suitable in CropSuite (red areas), these areas are less in SPAM2017 but considerably lower in SPAM2020. We included this comparison to the Supplement.

[Figure]

*Figure S2: As Figure 7b but comparison of CropSuite with (a) SPAM2000, (b) SPAM2017 , and (c) MapSPAM 2020, exemplarily for maize.*

5. Line 277, is it because that nutrient and soil fertility are not considered in this study?

**Reply:** We compared against GAEZ data with 'high input level', assuming that nutrients are not limiting for this option, as assumed also in CropSuite. For a better explanation, we added a more detailed description of these assumptions in L310-312.

We think that the differences mainly result due to differences between the different soil data used (HWSD in GAEZ, SoilGrids in CropSuite). This is illustrated by the following figure, indicating more gradual changes in CropSuite, whereas GAEZ shows strong and abrupt changes, especially between borders (e.g. between Angola and Zambia). This follows patterns of the underlying soil data, which is a known issue in the HWSD data.

[Figure]

*Suitability for maize for CropSuite (left) and GAEZv4 (right).*

6. In theory, I would expect a smaller area in this study because this study considers additional climate variability. However, Figure 8 shows a larger area by this study. Can the authors explain more about this?

**Reply:** Thank you, this aspect was not addressed adequately so far. To better illustrate the impact of climate variability in the comparison with GAEZ, we do not consider climate variability any more in Figure 8 (as we did previously). In addition to Fig. 8, we included Fig. S4 to the supplement, showing the comparison with the consideration of climate variability and discuss the difference in the results section 336-337. When considering climate variability in CropSuite, orange and grey bars increase as larger areas are considered unsuitable due to climate variability (see Fig. S4 compared to revised Fig. 8).

[Figure]

*Figure S4: Same as Fig. 8 but with consideration of climate variability for CropSuite.*

---

## Referee Report (RR1)

**Review 2: "CropSuite - A comprehensive open-source crop suitability model considering climate variability for climate impact assessment" (egusphere-2024-2526)**

**General comments**

Thanks to the authors for their revisions. I have a few more minor comments. (Line numbers in my notes refer to the tracked-changes version.)

Colors
- Re: the colorbars of Figs. 10(a) and 12(a, d)
  - There are indeed no points where a one-pixel change results in a categorical color difference, but there are some very sharp gradients. This blog post explains why this isn't great for figure design (even when not considering color vision deficiencies). In your case I actually think such gradients could be okay, but only if they (approximately) line up with the boundaries between your suitability bins (0/1, 32/33, 74/75).
  - Look at the difference between, e.g., 16 and 24, where it goes from gray to red. This stark difference contrasts with the fact that those are both categorized as "unsuitable" according to Table 3.
  - The "perceptually uniform sequential colormaps" at the "Choosing colormaps in Matplotlib" webpage are great choices without this issue that still work under red-blindness and green-blindness.
  - The GMD Guidelines for Authors section on Figures & Tables recommends strongly that figures should be made accessible to people with color vision deficiency. I'm not sure "other people use the same inaccessible color scheme" is a good enough reason to ignore that.
  - Also note that the FAO plot linked doesn't actually use red, but rather brown. So it's not the same color scale anyway. (Not that the FAO scale is any more colorblind-friendly.)
- If keeping some colorblind-friendly maps out of the main text, they should be included in the supplemental PDF, not in a separate 5 GB (!!) file. They should also be referenced in the captions of the figures in question.

Other
- Reviewer 2 had the following comment: "In theory, I would expect a smaller area in this study because this study considers additional climate variability. However, Figure 8 shows a larger area by this study. Can the authors explain more about this?" The authors changed Fig. 8 to not consider climate variability for consistency with GAEZ, which makes sense, and they note that when variability *is* considered, more

area is considered unsuitable (i.e., the purple bars shrink and orange bars grow between Figs. 8 and S4). However, the reviewer's original comment still stands: There are still a lot of crops where a substantial fraction of their CropSuite-suitable area is GAEZ-unsuitable. The Results or Discussion might benefit from highlighting this and perhaps investigating the reasons for one such crop (e.g., cabbage).

- Fig. 9: Some of the bars (e.g., rye) seem to have changed color (i.e., climatically suitable area value) pretty dramatically between the original manuscript and the revision. What happened there?
- Figs. 9 and S5 look identical to my eyes; please double-check that the correct figures were both included.
- Fig. S5 caption: "modulo" should be "mode." Sorry for the confusion in my original comment.
- Fig. 12a: Color bar label is only partially visible.
- Great job with the Fig. 13b redesign.
- Thank you for the response to my "Do there tend to be any patterns in the discrepancies that might explain them?" question. Please consider including something like that in the Results or Discussion (sorry if it's there and I missed it!).

---

## Author Response (AR2)

**Review 2: "CropSuite - A comprehensive open-source crop suitability model considering climate variability for climate impact assessment" (egusphere-2024-2526)**

**General comments**

Thanks to the authors for their revisions. I have a few more minor comments. (Line numbers in my notes refer to the tracked-changes version.)

Reply: Dear reviewer, we thank you your sincere interest and support! We think we have now been able to clarify the final details. Unfortunately, the links embedded into your comments were not available for us in the pdf. However, we have researched the respective entries ourselves and found good content (although possibly not what was originally intended).

Colors
- Re: the colorbars of Figs. 10(a) and 12(a, d)
  - There are indeed no points where a one-pixel change results in a categorical color difference, but there are some very sharp gradients. This blog post explains why this isn't great for figure design (even when not considering color vision deficiencies). In your case I actually think such gradients could be okay, but only if they (approximately) line up with the boundaries between your suitability bins (0/1, 32/33, 74/75).
  - Look at the difference between, e.g., 16 and 24, where it goes from gray to red. This stark difference contrasts with the fact that those are both categorized as "unsuitable" according to Table 3.
  - The "perceptually uniform sequential colormaps" at the "Choosing colormaps in Matplotlib" webpage are great choices without this issue that still work under red-blindness and green-blindness.
  - The GMD Guidelines for Authors section on Figures & Tables recommends strongly that figures should be made accessible to people with color vision deficiency. I'm not sure "other people use the same inaccessible color scheme" is a good enough reason to ignore that.
  - Also note that the FAO plot linked doesn't actually use red, but rather brown. So it's not the same color scale anyway. (Not that the FAO scale is any more colorblind-friendly.)
- If keeping some colorblind-friendly maps out of the main text, they should be included in the supplemental PDF , not in a separate 5 GB (!!) file. They should also be referenced in the captions of the figures in question.

Reply:
The crop suitability is a continuous dataset between 0 and 100 (actually between 0 and 1, however we save the data as 8-bit integer between 0 and 100 to increase efficiency and save disk space and memory). We added a short description in Line 63-65 to better explain why the data is saved between 0 and 100. The suitability categories are basically interchangeable and arbitrary. However, they are helpful and required to analyze and assess the results. The data itself, however, is not categorical or discrete. Therefore, a reduction or increase of suitability by e.g. 8 points has the same quantitative meaning, regardless of whether the difference falls into a different category or not.

You are right that GMD strongly encourages authors to use colorblind friendly colormaps. Therefore, we replaced Figs 10(a) and 12(a, b, d) in the main text with colorblind-friendly colormaps. Thanks a lot for your suggestions. We now chose 'viridis_r' in Matplotlib. We now added the figures with the more color-intuitive colormaps which are however not colorblind-friendly but are similarly used by other approaches to the supplement. We think that this is a good compromise, that is bringing all interests and arguments together. In addition, as already said, all maps are available for download, in both versions. In CropSuite (v1.0), we put a lot of effort into being user-friendly. For colorblind people, we also implemented the possibility to output standard maps in colorblind-friendly colormaps.

Other
- Reviewer 2 had the following comment: "In theory, I would expect a smaller area in this study because this study considers additional climate variability. However, Figure 8 shows a larger area by this study. Can the authors explain more about this?" The authors changed Fig. 8 to not consider climate variability for consistency with GAEZ, which makes sense, and they note that when variability is considered, more area is considered unsuitable (i.e., the purple bars shrink and orange bars grow between Figs. 8 and S4). However, the reviewer's original comment still stands: There are still a lot of crops where a substantial fraction of their CropSuite-suitable area is GAEZ-unsuitable. The Results or Discussion might benefit from highlighting this and perhaps investigating the reasons for one such crop (e.g., cabbage).

  Reply: We generally find more CropSuite area suitable than in GAEZ (with or without variability). We already discuss this issue, which we particularly identified for barley, cabbage, chickpea, rapeseed, rye

and wheat in Ln 333-336. This point is already also discussed in reply #5 to the other reviewer. We think that the different soil data used in GAEZ (HWSD) and CropSuite (SoilGrids) is a main reason of this difference. This is illustrated by the following figure, indicating more gradual changes in CropSuite, whereas GAEZ shows strong and abrupt changes, especially between borders (e.g. between Angola and Zambia). This follows patterns of the underlying soil data, which is a known issue in the HWSD data. We added a paragraph to ln338-340 to better address this issue!

[Figure]

*Suitability for maize for CropSuite (left) and GAEZv4 (right).*

- Fig. 9: Some of the bars (e.g., rye) seem to have changed color (i.e., climatically suitable area value) pretty dramatically between the original manuscript and the revision. What happened there?
  Reply: For rye (only), we identified an error in the climatically suitable area. This was corrected. For the other crops, nothing changed, but the colorbar limits changed between the two plots!

- Figs. 9 and S5 look identical to my eyes; please double-check that the correct figures were both included.
  Reply: Thanks a lot for taking such a close look! Indeed, they were identical. We inserted Fig. S5 as Fig. 5 by mistake. We apologize a lot for that mistake. We changed Fig. 5 accordingly with the correct one. Regarding the previous question, the colors of the bars are the same again than in the initial version (except for rye).

- Fig. S5 caption: "modulo" should be "mode. " Sorry for the confusion in my original comment.
  Reply: Thank you, we corrected that.

- Fig. 12a: Color bar label is only partially visible.
  Reply: Thank you, we corrected that.

- Great job with the Fig. 13b redesign.
  Reply: Thanks! It was some work.

  - Thank you for the response to my "Do there tend to be any patterns in the discrepancies that might explain them?" question. Please consider including something like that in the Results or Discussion (sorry if it's there and I missed it!).
  Reply: We added a paragraph on this in line 357-362 and also added Fig. S6 to the Supplement, showing the histogram for comparison between CropSuite and the GGCMI crop calendar.